# Search for *Truth* from Reasoning: A Dynamic Representation Editing Framework for Steering LLM Trajectories

**Tianlong Wang** [1 2 3 *] **Yuhang Wang** [1 2 3 *] **Weibin Liao** [1 2 3 * ‡]
**Xin Gao** [1 2 3] **Xinyu Ma** [1 2 3] **Yang Lin** [1 2 3] **Yasha Wang** [1 3 4 ✉] **Liantao Ma** [1 2 3 ✉]

## Abstract

Current approaches to enhance Large Language Model (LLM) reasoning, such as Chain-of-Thought and "Wait" prompts, primarily encourage models to *think more*, yet often fail to guide them toward *Truth*. While Representation Editing (RepE) offers a intrinsic control, its application to dynamic reasoning trajectories remains underexplored. In this work, we bridge this gap by investigating the geometry of truth within unfolding reasoning chains. We uncover three critical insights: (1) *Truth* is encoded at the sentence level and is entangled with latent reasoning patterns; (2) Effective intervention follows an *Uncertainty Principle* and a *Decay Effect*, requiring localization to early, high-entropy forks; (3) Naive steering vectors suffer from noise, risking collateral damage to correct trajectories. Based on these findings, we propose `DynaSteer`, a dynamic RepE framework. `DynaSteer` employs pattern clustering to disentangle reasoning manifolds and utilizes Fisher-LDA to project purified truth. By dynamically monitoring lookahead entropy, it selectively steers and rolls back trajectories only when necessary. Comprehensive experimental results on several MATH benchmark verify the effectiveness of `DynaSteer`, and experiments on out-of-domain coding tasks further confirm its generalization ability. Our code is publicly available at https://github.com/tianlwang/DynaSteer.

*Equal contribution ‡ Project Lead [1]National Engineering Research Center of Software Engineering, Peking University [2]School of Computer Science, Peking University [3]Key Laboratory of High Confidence Software Technologies, Ministry of Education [4]Peking University Information Technology Institute (Tianjin Binhai). Correspondence to: Tianlong Wang <tianlong.wang@stu.pku.edu.cn>, Weibin Liao <liaoweibin@stu.pku.edu.cn>, Liantao Ma <malt@pku.edu.cn>.

*Proceedings of the 43rd International Conference on Machine Learning*, Seoul, South Korea. PMLR 306, 2026. Copyright 2026 by the author(s).

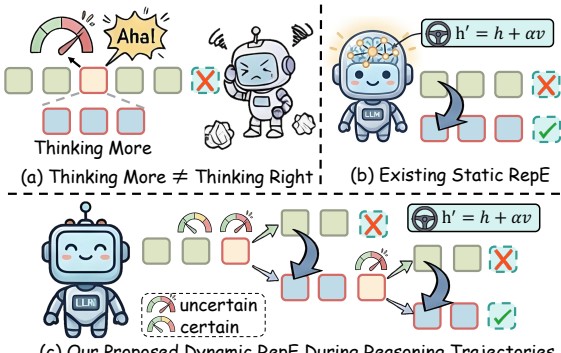

*Figure 1.* Illustration of the key differences between (a) existing reasoning intervention approaches, (b) static representation editing and (c) our dynamic representation editing during reasoning trajectories.

## 1. Introduction

Large Language Models (LLMs) have demonstrated remarkable proficiency in complex reasoning tasks (Huang & Chang, 2023; Plaat et al., 2024; Wang et al., 2025b), largely driven by the Chain-of-Thought (CoT) paradigm (Wei et al., 2022; Yao et al., 2023; Ding et al., 2025b; Fang et al., 2026a) which elicits structured reasoning trajectories. Recent advancements (Yang et al., 2025; Zhou et al., 2025; Cai et al., 2015; Pipis et al., 2025) have further attempted to enhance this capability through inspired prompting strategies, such as the "Aha moment" or "Wait" prompts. While these methods successfully encourage LLMs to generate more tokens and engage in deeper contemplation, they primarily induce the model to **think more** without explicitly **guiding it to steer** toward correct deductions among bifurcating possibilities. Consequently, when LLMs face uncertainty, simply extending the generation length often leads to "hallucination snowballs" (Zhong et al., 2024) rather than self-correction.

To address this limitation, **Representation Editing (RepE)** (Li et al., 2023; Wang et al., 2025c; Ma et al., 2025) has emerged as a promising avenue for intrinsic control. Unlike prompting, RepE directly pinpoints and modulates the internal activation patterns of LLMs, offering a more fundamental remedy by guiding the hidden states toward a target concept. However, a critical gap remains in the litera-

ture. While RepE has achieved notable success in aligning **static** behavioral attributes, such as language style (Ma et al., 2025; Shen et al., 2025), honesty (Xiao et al., 2025), and harmlessness (Dong et al., 2024), its application to the **dynamic reasoning process** remains largely underexplored. Prior works (Li et al., 2023; Wang et al., 2025c) predominantly focus on non-reasoning modes or contexts where the conclusion is explicitly visible. The challenge of steering the model towards *Truth* during the intermediate derivation stage, *before* a final answer is reached, presents unique spatiotemporal complexities.

*"**Truth**, a binary, geometrically encoded directional signal within the LLM's activation space, whose sign discriminates whether an ongoing reasoning trajectory ultimately arrives at a verifiable correct or incorrect answer."*

In this work, we conduct a **systematic exploration** of applying RepE to guide LLM reasoning. Our investigation is structured around three pivotal research questions, *Existence*, *Locus*, and *Mitigation*, that bridge the gap between static attribute editing and dynamic reasoning guidance:

1. *Existence of Truth*: **Does the *Truth* exist within LLM representations along reasoning trajectories?** A fundamental prerequisite for RepE is that internal representations must distinguish between *Truth* and *Fallacy*. Our probing experiments reveal a crucial nuance: naive token-level probes struggle to discern *Truth*, suggesting it is not encoded in atomic units. However, when elevated to the sentence level, the latent distinction becomes evident. We further discover that the *Truth Steering* is inherently **entangled** with specific reasoning patterns. By disentangling these patterns via clustering, we observe a significant enhancement in discrimination accuracy.

2. *Locus of Intervention*: **Where and when is intervention effective?** We identify two critical spatiotemporal dynamics that govern effective steering. First, the *Decay Effect*: intervention is most potent in the early stages of reasoning; as the chain lengthens, the model becomes "entrenched" in its trajectory, rendering late-stage editing ineffective. Second, the *Uncertainty Principle*: effective intervention strictly correlates with high entropy. Editing is necessary only at "reasoning forks" where the model exhibits uncertainty; intervening at low-entropy positions yields negligible benefits or disrupts coherence.

3. *Mitigation of Risk*: **How to mitigate the potential risks associated with intervention?** Standard RepE methods typically derive steering vectors using a naive Mean-Difference approach. However, given the entanglement of *Truth* with other high-level semantic concepts, this approach inevitably incorporates noise, posing a risk of corrupting originally correct trajectories. To mitigate this, we propose the use of **Fisher Linear Discriminant**

**Analysis (Fisher-LDA)**. By projecting representations onto a purified subspace, we maximize the separation of *Truth* while minimizing interference with the model's semantic manifold.

Based on these insights, we propose `DynaSteer`. `DynaSteer` is a dynamic inference-time RepE framework designed to steer reasoning trajectories proactively. It operates by (1) detecting critical reasoning forks via sentence-level lookahead entropy; (2) calculating purified *Truth Steering* using Fisher-LDA within disentangled reasoning clusters; and (3) applying a temporally decaying injection to guide the model's trajectory while employing a rollback mechanism to correct errors at their inception.

In summary, our main contributions are as follows:

- **Systematic Analysis of Dynamic RepE:** We provide the first comprehensive analysis bridging static attribute editing and dynamic reasoning. We reveal the *Decay Effect* and the *Uncertainty Principle*, establishing that intervention must be localized to early, high-entropy reasoning forks to be effective.

- **Subspace-Projected Steering Mechanism:** We identify that the geometric entanglement of truth and reasoning patterns limits standard RepE. We propose a novel application of Fisher-LDA to project internal representations onto a discriminative subspace, ensuring safe and precise steering.

- **`DynaSteer` Framework:** Synthesizing these findings, we introduce `DynaSteer`, an adaptive framework that dynamically monitors uncertainty and intervenes only when necessary. Extensive evaluations conducted in both in-domain and out-of-domain scenarios show that our approach substantially outperforms existing baselines on complex reasoning benchmarks.

## 2. Preliminary: Representation Editing

**Notation.** We consider an autoregressive Large Language Model (LLM) $\mathcal{M}$ with $L$ layers and $N$ heads. For an input sequence $x$, let $\mathbf{h}^{(l,n)} \in \mathbb{R}^d$ denote the internal activation at layer $l$ and head $n$, where $d$ is the hidden dimension. The model generates the next token based on the final layer representation via a decoding head. We denote a contrastive dataset as $\mathcal{D} = \{(x_i^+, x_i^-)\}_{i=1}^N$, where $x^+$ and $x^-$ correspond to inputs eliciting the target concept (e.g., truth) and its opposite (e.g., hallucination), respectively.

**Core Insight.** Representation Editing (RepE) operates on the geometry of the activation space. Its fundamental justification is the **Linear Representation Hypothesis** (Wang et al., 2025c; Li et al., 2023):

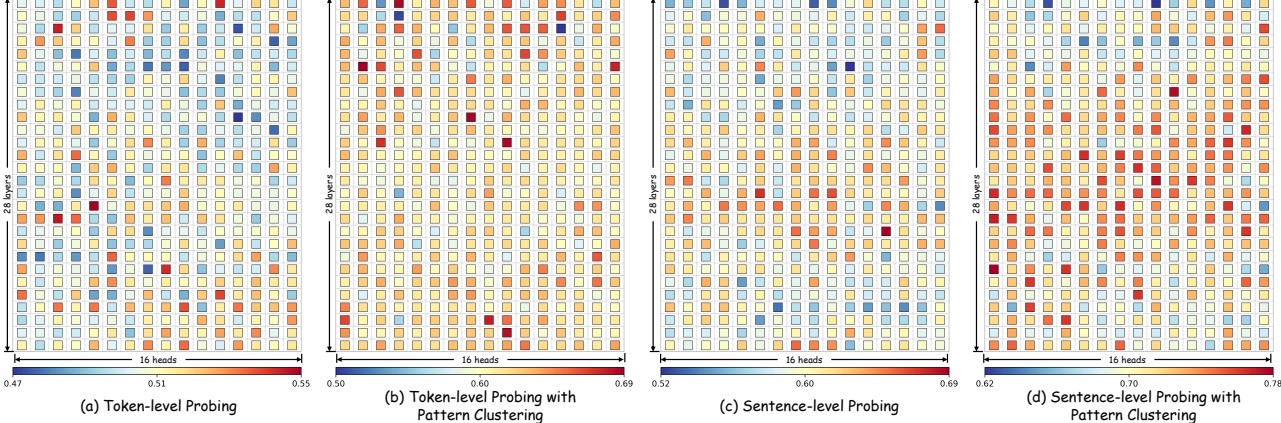

*Figure 2. Truth* discrimination accuracy across individual attention heads of the large language model. Lower rows correspond to shallower layers of the LLM. Expanding token-level representations to the sentence-level and incorporating pattern clustering substantially improve the accuracy of the linear probes.

> *RepE assumes that high-level semantic concepts are linearly encoded in the activation space. Specifically, there exists a direction vector $\mathbf{v} \in \mathbb{R}^d$ such that the intensity of the concept in any representation $\mathbf{h}$ is determined by the projection $\mathbf{h}^\top \mathbf{v}$.*

Based on this assumption, the steering vector $\mathbf{v}$ representing the target concept is typically extracted by computing the difference in means between positive and negative activations:

$$\mathbf{v} \approx \mathbb{E}_{x \sim \mathcal{D}}[\mathbf{h}(x^+)] - \mathbb{E}_{x \sim \mathcal{D}}[\mathbf{h}(x^-)]. \quad (1)$$

To control the model's behavior during inference, we intervene on the hidden state $\mathbf{h}$ by injecting this vector with a steering coefficient $\alpha$:

$$\tilde{\mathbf{h}} = \mathbf{h} + \alpha \cdot \frac{\mathbf{v}}{\|\mathbf{v}\|_2}. \quad (2)$$

This linear shift biases the model's subsequent computations towards the target concept without altering the model weights.

## 3. Does the *Truth* Exist Within LLM Representations Along Reasoning Trajectories?

The theoretical foundation of RepE rests on the **Linear Representation Hypothesis** (see Sec. 2). While prior work (Li et al., 2023; Wang et al., 2025c; Liao et al., 2026b; Zhang et al., 2025) has demonstrated that *Truth* can be encoded in representations containing the answer text, it remains unclear whether *Truth* is also present in reasoning trajectories where the answer is not observable.

Formally, let $\mathcal{M}$ denote an autoregressive LLM with $L$ layers and $H$ heads. For a given input sequence, let $\mathbf{h}_t^{(l,n)} \in \mathbb{R}^d$ denote the hidden-state activation at layer $l \in \{1, \dots, L\}$,

head $n \in \{1, \dots, N\}$, and token position $t$. We investigate whether there exists a global steering vector $\mathbf{v} \in \mathbb{R}^d$ such that the induced binary decision $\mathrm{sgn}\left(\mathbf{h}_t^{(l,n)\top} \mathbf{v}\right)$ serves as a reliable discriminator of the correctness of the trajectory. If such a vector exists, the sign of $\mathbf{h}_t^{(l,n)\top} \mathbf{v}$ should correlate systematically with the binary ground-truth label $y \in \{+1, -1\}$, where $y = +1$ denotes a trajectory leading to *Truth* and $y = -1$ denotes one leading to *Fallacy*. We deliberately employ linear probes to test the Linear Representation Hypothesis. By restricting the probe to minimal capacity, we ensure that the observed discriminability reflects the genuine linear structure of the *Truth* signal rather than the probe's own expressive power, adhering to the probe complexity control principle (Hewitt & Liang, 2019; Ding et al., 2025a).

**Semantic Granularity: From Tokens to Sentences.** Our initial probing experiments focused on the atomic unit of generation: the token. We defined the representation space at the token level, treating each $\mathbf{h}_t^{(l,n)}$ as an independent sample. In both token-level and sentence-level settings, the hidden state is extracted from the last token's activation of the input sequence. For token-level probing, the sequence is truncated at any arbitrary position. However, as shown in Fig. 2(a), our empirical results indicate that the linear separability of *Truth* at this granularity is poor, with classification accuracy saturating near 55%. We further elevate the analysis to the *sentence level*, where trajectory segmentation is guided by punctuation (e.g., . \n), restricting truncation exclusively to sentence boundaries. This acts as a filtering mechanism, ensuring hidden states are extracted only where complete semantic units form. As shown in Fig. 2(b), by shifting the domain from instantaneous token activations to semantically complete sentence representations, we observe a substantial increase in linear separability, with probing accuracy exceeding 69% (16%↑) across multiple attention

heads. This suggests that the *Truth* is an emergent property of semantic units rather than a localized feature of individual tokens.

**Manifold Disentanglement via Pattern Clustering.** Despite the improvements yielded by sentence-level aggregation, the decision boundary between *Truth* and *Fallacy* remains noisy. We hypothesize that this noise arises from *reasoning heterogeneity*: the geometric realization of *Truth* may vary depending on the underlying reasoning pattern (Bogdan et al., 2025; Liao et al., 2025) (e.g., plan generation, fact retrieval). If the manifold of reasoning representations is non-convex or consists of multiple disjoint regions, a single global linear probe will be suboptimal. To test this, we employ an unsupervised clustering approach to partition the activation space. Let $\mathcal{H} = \{\mathbf{h}_{S_i}\}_{i=1}^{N}$ be the set of collected sentence representations. We apply $k$-means clustering to decompose $\mathcal{H}$ into $M$ disjoint clusters $\mathcal{C}_1, \ldots, \mathcal{C}_M$, where each cluster conceptually corresponds to a distinct latent reasoning pattern. Within each cluster $\mathcal{C}_j$, we independently optimize a local linear probe defined by a cluster-specific *Truth Steering* $\mathbf{v}_j$.

$$\max_{\mathbf{v}_j} \sum_{\mathbf{h} \in \mathcal{C}_j} \mathbb{I}[\operatorname{sgn}(\mathbf{h}^\top \mathbf{v}_j) = y], \qquad (3)$$

where $\mathbb{I}[\cdot]$ is the indicator function. As shown in Fig. 2(b) and (d), post-clustering analysis reveals a dramatic improvement in separability, with intra-cluster classification accuracy reaching approximately 78% (23%↑). We further illustrate the separability of these reasoning patterns using a t-SNE visualization in Fig. 7. This finding implies that while the *Truth* signal is strong, it is *entangled* with the pattern of reasoning. The global geometry of *Truth* is likely piecewise linear, becoming linearly accessible only when the latent reasoning patterns are disentangled.

**Summary.** Our investigation confirms the existence of a recoverable *Truth* signal, subject to specific structural constraints.

> *Takeaway: We validate the core prerequisite for RepE: a discernible Truth Direction exists within the LLM activation space. However, this direction is not encoded in isolated token-level activations but emerges in sentence-level aggregates. Furthermore, the linearity of this signal is significantly enhanced when latent reasoning patterns are disentangled via clustering.*

## 4. Where and When Is Intervention Effective?

Having established the existence of the *Truth*, we now turn to the optimization of the intervention policy. A naive application of RepE at every time step is computationally prohibitive and risks degrading generation quality. Therefore, we seek to identify the optimal spatiotemporal coordinates

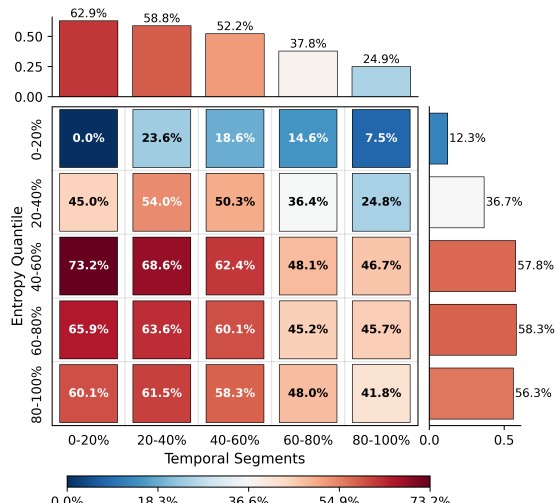

*Figure 3.* The relationship between the *Recovery Rate* of erroneous reasoning trajectories under representation editing and *Entropy Quantiles* across different *Temporal Segments*.

for intervention. We analyze this along two orthogonal axes: the **spatial locus** (identifying critical decision points) and the **temporal locus** (determining the effective window of opportunity).

**Spatial Locus: Entropy-Based Selection.** Reasoning is not a uniform process; it consists of deterministic transitions (e.g., syntactic completion) and stochastic decision points, which we term *reasoning forks*. We hypothesize that intervention is only necessary at these forks, where the model's internal confidence wavers between competing logical paths. To quantify this, we utilize the Shannon entropy (Yin et al., 2024) of the next-token distribution. Let $P(x_{t+1} \mid x_{\leq t})$ denote the probability distribution over the vocabulary $\mathcal{V}$ at time step $t$. The local uncertainty is given by:

$$\mathcal{H}(t) = -\sum_{w \in \mathcal{V}} P(w \mid x_{\leq t}) \log P(w \mid x_{\leq t}). \qquad (4)$$

Previous works (Wang et al., 2025a; Hao et al., 2025) observe that peaks in $\mathcal{H}(t)$ typically align with the initiation of new logical steps. Based on this, we investigate a **Selective Editing** strategy. Let $\tau$ be a dynamic threshold derived from the distribution of entropy values within a trajectory (e.g., the top-$k$ percentile). The intervention set is defined as $\mathcal{I}_{\text{select}} = \{t \mid H(t) > \tau\}$.

We investigate the relationship between the intrinsic *Recovery Rate* and entropy quantiles. Let $\mathcal{S}_{neg}$ denote the set of sentence-level segments belonging to trajectories that originally result in incorrect answers ($y = -1$). For a given segment $s \in \mathcal{S}_{neg}$, we generate $N$ independent stochastic rollouts. Let $\{y_1, y_2, \ldots, y_N\}$ denote the binary correctness labels of the resulting trajectories for these rollouts. We define the segment $s$ as *recoverable* if the correct answer exists within the sampled space. Formally, the recoverability

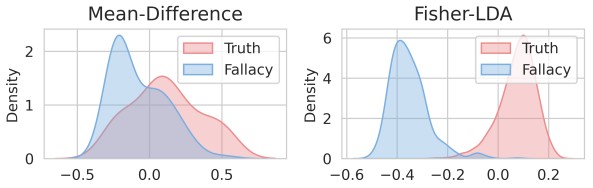

*Figure 4.* Distributional comparison of *Truth* and *Fallacy* representations under Mean-Difference and Fisher LDA projections.

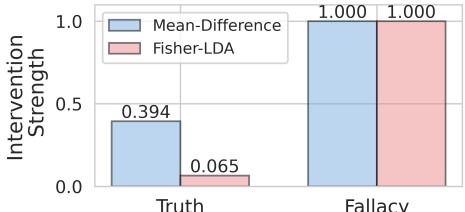

*Figure 5.* Intervention strength of representation editing on *Truth* and *Fallacy* under Mean-Difference and Fisher-LDA.

indicator $\mathcal{R}(s)$ is defined as:

$$\mathcal{R}(s) = \mathbb{I}\left[\exists n \in \{1, \dots, N\} : y_n = 1\right], \tag{5}$$

where $\mathbb{I}[\cdot]$ is the indicator function. The *Recovery Rate* for a group of segments is then calculated as the expected value of $\mathcal{R}(s)$.

As illustrated in Fig. 3, we observe a distinct correlation: for segments with low entropy, the *Recovery Rate* is minimal (peaking at only 12.3%), indicating that the model is deeply "entrenched" in the fallacy and consistently reproduces the error even under stochastic sampling. Conversely, at high-entropy forks, the *Recovery Rate* rises significantly to 58.3%. This suggests that at these uncertain moments, the *Truth* is still accessible within the model's hypothesis space. Thus, entropy serves as a highly effective proxy for identifying the spatial locus of intervention.

**Temporal Locus: The Decay Effect.** Beyond spatial selection, the timing of intervention relative to the overall trajectory is critical. We investigate the *temporal sensitivity* of the model's intrinsic error-correction capability. We divide the reasoning trajectories into 5 relative temporal segments $T_1, \dots, T_5$. Our experiments, as shown in Fig. 3, reveal a pronounced monotonic decay in this intrinsic potential: $\mathcal{R}(T_1) \gg \mathcal{R}(T_5)$. Specifically, segments located in the early stage ($T_1$) exhibit a hig *Recovery Rate* of 62.9%, whereas those in the final stage ($T_5$) drop significantly to 24.9%.

We term this phenomenon the *Decay Effect*. It suggests that autoregressive reasoning exhibits strong path dependence. As the model generates more tokens consistent with an initial fallacy, its internal state becomes "entrenched" in that error. The hidden states $\mathbf{h}_t$ at later time steps $t \gg 1$ are conditioned on a long history of incorrect tokens $x_{<t}$, making

the incorrect trajectory a deep local minimum in the energy landscape. This implies that the "window of opportunity" for effective steering is transient. Interventions must be *anticipatory*: corrections are most feasible at the inception of a reasoning error ($T_1$), before the trajectory collapses into a deep local minimum from which even stochastic exploration cannot escape.

**Summary.** The integration of spatial and temporal analyses defines a precise regime for efficient RepE.

> *Takeaway: Effective intervention is highly localized in both space and time. Spatially, editing is strictly necessary only at high-entropy reasoning forks. Temporally, intervention must occur early in the derivation process, as the model's susceptibility to correction decays rapidly due to the entrenchment of logical errors over time.*

## 5. How to Mitigate the Potential Risks Associated with Intervention?

While the previous sections established the existence of *Truth* and the optimal timing for its application, a critical safety challenge remains. As shown in Fig. 4, the distributions of representations for *Truth* ($\mathbf{h}^+$) and *Fallacy* ($\mathbf{h}^-$) are not perfectly disjoint. This overlap creates the risk of *False Positive Intervention*, where the steering vector inadvertently perturbs a correct reasoning process towards *Fallacy*. In this section, we analyze this risk through a geometric lens and propose a subspace projection method to mitigate it.

**The Risk of Naive Mean-Difference Steering.** Standard RepE typically estimates the steering direction using the difference of class means. Let $\mathcal{D}^+$ and $\mathcal{D}^-$ be the sets of hidden representations for *Truth* and *Fallacy*, respectively. The Mean-Difference (MD) vector is defined as:

$$\mathbf{v}_{\text{MD}} = \boldsymbol{\mu}^+ - \boldsymbol{\mu}^-, \tag{6}$$

$$\boldsymbol{\mu}^c = \frac{1}{|\mathcal{D}^c|} \sum_{\mathbf{h} \in \mathcal{D}^c} \mathbf{h}, c \in \{+, -\}. \tag{7}$$

The intervention strength at inference time is proportional to the projection scalar $\mathbf{h}^\top \mathbf{v}_{\text{MD}}$. Ideally, this projection should act as a robust discriminator: maximizing intensity for *Fallacy* while vanishing (or becoming negative) for *Truth*.

However, the density plots in Fig. 4 reveal that upon projecting $h$ onto $\mathbf{v}_{\text{MD}}$, a significant distributional overlap emerges, where a portion of *Truth* samples is indistinguishable from *Fallacy*. This entanglement is further quantified in Fig. 5: when the intervention strength for *Fallacy* is normalized to 1.000, *Truth* samples inadvertently inherit a substantial strength of 0.394. This makes naive RepE a *losing proposition*: the expected utility is negative because the harm

caused by corrupting *Truth* trajectories outweighs the benefit of repairing *Fallacy* ones.

**Fisher-LDA for *Truth* Purification.** We attribute the failure of naive mean-difference steering to the extreme **anisotropy** of the activation space, where non-informative outlier dimensions dominate the variance (Rudman & Eickhoff, 2024). To purify the *Truth* signal, we employ Fisher Linear Discriminant Analysis (Fisher-LDA) to find a direction that maximizes class separability while suppressing within-class noise.

Let $\boldsymbol{\delta} = \boldsymbol{\mu}^+ - \boldsymbol{\mu}^-$ be the mean difference between *Truth* and *Fallacy* representations. We define the within-class covariance matrix $\boldsymbol{\Sigma}_w$ as:

$$\boldsymbol{\Sigma}_w = \sum_{c \in \{+,-\}} \frac{1}{|\mathcal{D}_c|} \sum_{\mathbf{h} \in \mathcal{D}^c} (\mathbf{h} - \boldsymbol{\mu}^c)(\mathbf{h} - \boldsymbol{\mu}^c)^\top. \quad (8)$$

The optimal steering axis $\mathbf{w}^*$ is derived by maximizing the Rayleigh Quotient (Rayleigh, 2013), yielding the closed-form solution $\mathbf{w}^* = \boldsymbol{\Sigma}_w^{-1} \boldsymbol{\delta}$ (see Appendix B for the detailed derivation). Under the Fisher-LDA paradigm, the adaptive intervention intensity during inference is computed as:

$$\eta(\mathbf{h}) = \alpha \cdot \tanh\left(\mathrm{ReLU}\left(\frac{\mu_{p\_proj} - h_{proj}}{\sigma_{p\_proj}}\right)\right), \quad (9)$$

where $h_{proj} = \mathbf{w}^{*\top}\mathbf{h}$, and $\mu_{p\_proj}, \sigma_{p\_proj}$ are the mean and standard deviation of positive samples in the projected space. This formulation acts as an asymmetric gate: the *ReLU* ensures that intervention is selectively applied only to representations lagging behind the *Truth* center, while the Z-score normalization and *tanh* function provide scale-invariant alignment and numerical stability against outliers. Finally, $\alpha$ serves as a hyperparameter that regulates the global magnitude of the intervention.

As illustrated in Fig. 4 and Fig. 5, projecting $h$ onto the purified axis $\mathbf{w}^*$ significantly reduces the distributional overlap compared to the naive mean-difference. This ensures that *Truth* representations receive minimal perturbation, thereby transforming RepE from a risky heuristic into a controlled and safe mechanism.

**Summary.** We address the geometric entanglement of truth and semantics to ensure safe intervention.

> *Takeaway: Naive mean-difference steering risks corrupting correct reasoning due to significant distributional overlap. By employing Fisher-LDA, we identify a purified subspace that maximizes the signal-to-noise ratio of the truth direction. This significantly reduces collateral damage to correct trajectories, ensuring a positive cost–benefit trade-off for the intervention.*

# 6. Methodology

Building upon the empirical evidence of a recoverable *Truth* signal (Sec. 3) and the spatiotemporal constraints of intervention (Sec. 4), we introduce `DynaSteer`, a framework that bridges the gap between static representation editing and dynamic reasoning trajectories.

## 6.1. Consistency-based *Truth* Annotation

To align the representation space with semantic reasoning steps rather than atomic tokens, `DynaSteer` operates at the level of trajectory sentences. Let a reasoning trajectory generated by the model $\mathcal{M}$ be decomposed into a sequence of sentences $\mathcal{T} = (S_1, S_2, \ldots, S_T)$, where each sentence $S_t$ represents a coherent semantic unit, such as a clause or a deduction step.

`DynaSteer` employs a consistency-based filtering strategy to construct a robust dataset $\mathcal{D}$. For a given sentence $S_t$ located at a high-entropy reasoning fork, the framework performs $N$ stochastic rollouts to generate a set of final answers $\{y_1, \ldots, y_N\}$. The semantic label $L(S_t)$ for the hidden representations associated with this sentence is assigned via consensus:

$$L(S_t) = \begin{cases} +1(\text{\textit{Truth}}) & \text{if } y_n = 1 \forall n \in \{1, \ldots, N\}, \\ -1(\text{\textit{Fallacy}}) & \text{if } y_n \neq 1 \forall n \in \{1, \ldots, N\}, \\ \perp(\text{Discard}) & \text{otherwise.} \end{cases}$$

$$(10)$$

Subsequently, `DynaSteer` collects the internal hidden states $\mathbf{h} \in \mathbb{R}^d$ corresponding to accepted sentences $(L(S_t) \neq \perp)$ to form the paired dataset $\mathcal{D} = \{(\mathbf{h}_i, L(S_i))\}$.

## 6.2. *Truth*-Relevant Head Location

Not all attention heads encode the *Truth* signal equally. To ensure computational efficiency and intervention precision, `DynaSteer` identifies the specific subset of heads that act as the locus for truth differentiation. Using the labeled dataset $\mathcal{D}$, it trains linear probes on the hidden states of each head across all layers.

`DynaSteer` ranks the heads based on their probing accuracy in distinguishing *Truth* $(+1)$ from *Fallacy* $(-1)$ and selects the top-$k$ heads, denoted as $\mathcal{H}_{\text{top}}$, to serve as the intervention targets. Consequently, the subsequent clustering and steering operations are restricted exclusively to representations $\mathbf{h}^{(l,h)}$ where $(l, h) \in \mathcal{H}_{\text{top}}$.

## 6.3. Manifold Disentanglement & Truth Purification

As established in Sec. 3, the geometry of *Truth* is entangled with latent reasoning patterns. `DynaSteer` addresses this via a two-stage offline process: separating reasoning modes and then purifying the truth direction within those modes.

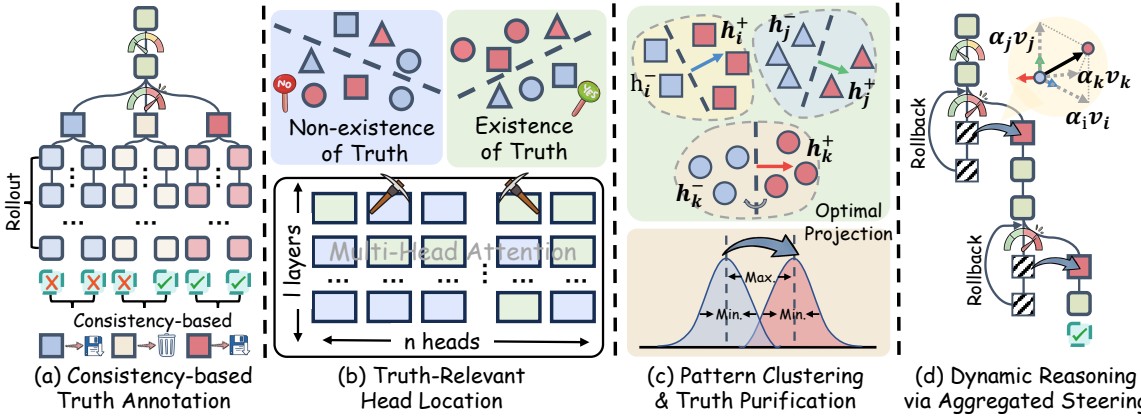

(a) Consistency-based Truth Annotation  (b) Truth-Relevant Head Location  (c) Pattern Clustering & Truth Purification  (d) Dynamic Reasoning via Aggregated Steering

*Figure 6.* Overview of the `DynaSteer` framework. (a) illustrates the data construction phase, where `DynaSteer` samples high-entropy sentences and uses consistency to distinguish between *Truth* and *Fallacy*. (b) presents the use of probing techniques to localize *Truth*-relevant attention heads within the LLM. In (c), `DynaSteer` utilizes pattern clustering for manifold disentanglement, followed by Fisher-LDA to derive pure *Truth Steering*. Finally, (d) shows the dynamic inference stage, where lookahead entropy monitoring identifies trajectories that need correction, applying an adaptive aggregated steering to guide the reasoning.

**Manifold Disentanglement via Pattern Clustering.** `DynaSteer` hypothesizes that the activation space consists of multiple reasoning manifolds. To disentangle these, it computes the raw difference vector $\boldsymbol{\delta}_i = \mathbf{h}_i^+ - \mathbf{h}_i^-$ for pairs of positive and negative forks derived from the same context. `DynaSteer` applies $k$-means clustering to the set of difference vectors $\{\boldsymbol{\delta}_i\}$, partitioning the space into $M$ disjoint clusters $\{\mathcal{C}_1, \ldots, \mathcal{C}_M\}$. Each cluster $\mathcal{C}_m$ conceptually represents a distinct reasoning pattern (e.g., symbolic logic vs. arithmetic derivation).

***Truth* Purification via Fisher-LDA.** Within each reasoning cluster $\mathcal{C}_m$, to extract a purified steering direction, `DynaSteer` employs Fisher-LDA (Sec. 5).

For a specific cluster $\mathcal{C}_m$ and head, let $\boldsymbol{\mu}_m^+$ and $\boldsymbol{\mu}_m^-$ denote the centroids of the *Truth* and *Fallacy* classes, respectively. `DynaSteer` computes the within-class scatter matrix $\Sigma_{w,m}$ and calculates steering vector $\mathbf{v}_m$:

$$\mathbf{v}_m = \Sigma_{w,m}^{-1}(\boldsymbol{\mu}_m^+ - \boldsymbol{\mu}_m^-). \quad (11)$$

The vector $\mathbf{v}_m$ defines the normal to the hyperplane that best discriminates truth from fallacy within the specific reasoning mode $m$.

### 6.4. Dynamic Reasoning via Aggregated Steering

During the online inference phase, `DynaSteer` applies the pre-computed steering vectors dynamically. This process is governed by entropy monitoring and a rollback mechanism to handle the path-dependent nature of generation.

**Lookahead Entropy Monitoring.** To avoid unnecessary perturbations, `DynaSteer` adopts a selective intervention strategy. As the model generates a candidate sentence $S_{\text{cand}}$, the framework computes its average token entropy $H(S_{\text{cand}})$. It employs a threshold $\gamma$, set to the $80^{\text{th}}$ percentile of the

training entropy distribution, following prior work (Wang et al., 2025a).

- If $H(S_{\text{cand}}) \leq \gamma$, the sentence is deemed deterministic and reliable; generation proceeds normally.

- If $H(S_{\text{cand}}) > \gamma$, the sentence represents a high-uncertainty reasoning fork, triggering the intervention mechanism.

**Temporal Decay and Dynamic Strength Calculation.** When intervention is triggered at step $t$, `DynaSteer` calculates the steering parameters. First, to account for the *Decay Effect*, we define a temporal gating coefficient $\beta(t) = \mathbb{I}[t \leq \tau_{\text{decay}}]$, where $\tau_{\text{decay}}$ is a pre-defined threshold. Intervention is applied only when $\beta(t) = 1$. Second, `DynaSteer` computes the dynamic intervention strength $\eta_m(\mathbf{h})$ for each reasoning cluster $\mathcal{C}_m$ according to Eq. 9.

**Multi-Steering Aggregation and Rollback.** The final steering is an aggregate of the purified directions from all reasoning clusters. The edited hidden state $\tilde{\mathbf{h}}$ is formulated as:

$$\tilde{\mathbf{h}} = \mathbf{h} + \beta(t)\sum_{m=1}^{M} \eta_m(\mathbf{h})\mathbf{v}_m. \quad (12)$$

Crucially, mere injection is insufficient if the candidate sentence $S_{\text{cand}}$ has already committed to an error. Therefore, `DynaSteer` executes a **Rollback**: it discards $S_{\text{cand}}$ and forces the model to traverse the reasoning fork again, now guided by the purified *Truth* signal.

## 7. Experiment

**Experimental Setups.** We present our experimental setup in Appendix. E, including the network architecture, datasets, evaluation metrics, baselines, and implementation details.

*Table 1.* Performance comparison of Original, "Wait" prompt, RepE methods, and DynaSteer across three backbone LLMs on three MATH datasets and two Coding datasets. The best results are highlighted in **bold**, while the second-best results are underlined. The *Impro* (%) indicates the relative improvement of DynaSteer over the second-best performance. ID: In-Domain. OOD: Out-of-Domain.

| Methods | MATH (ID) | | | Coding (OOD) | |
|---|---|---|---|---|---|
| | GSM8K | MATH500 | AMC23 | HumanEval+ | MBPP+ |
| *Qwen3-1.7B* | | | | | |
| Plain | 79.61 | 69.20 | 40.00 | 65.24 | 50.26 |
| "Wait" | 81.05 | 71.20 | 40.00 | 65.85 | 50.26 |
| ITI | 81.50 | 71.80 | 37.50 | 65.24 | 50.00 |
| ACT | 81.73 | 72.60 | 42.50 | 65.85 | 51.06 |
| DRESS | 81.65 | 72.80 | 42.50 | 65.24 | 51.32 |
| DynaSteer | **82.87** | **74.80** | **47.50** | **67.68** | **54.50** |
| *Impro.(%)* | 1.39 | 2.75 | 11.76 | 2.78 | 6.20 |
| *Llama3.2-3B-Instruct* | | | | | |
| Plain | 80.14 | 47.80 | 27.50 | 47.56 | 50.53 |
| "Wait" | 80.06 | 46.20 | 25.00 | 48.17 | 50.53 |
| ITI | 79.98 | 48.40 | 25.00 | 48.78 | 51.06 |
| ACT | 80.29 | 48.80 | 32.50 | 51.83 | 52.65 |
| DRESS | 80.06 | 48.60 | 30.00 | 51.22 | 51.85 |
| DynaSteer | **81.35** | **52.40** | **35.00** | **54.27** | **54.50** |
| *Impro.(%)* | 1.32 | 7.38 | 7.69 | 4.71 | 3.51 |
| *Qwen3-8B* | | | | | |
| Plain | 92.49 | 84.00 | 65.00 | 79.88 | 61.90 |
| "Wait" | 92.57 | 84.20 | 65.00 | 78.66 | 61.64 |
| ITI | 92.27 | 84.20 | 67.50 | 79.27 | 62.17 |
| ACT | 92.80 | 84.80 | 70.00 | 79.88 | 62.43 |
| DRESS | 92.72 | 84.60 | 67.50 | 80.49 | 63.76 |
| DynaSteer | **94.39** | **86.20** | **72.50** | **81.71** | **65.61** |
| *Impro.(%)* | 1.71 | 1.65 | 3.57 | 1.52 | 2.90 |

*Table 2.* Ablation Studies of DynaSteer using Qwen3-1.7B as backbone LLM on MATH and Coding tasks. ♣ represents that the advantage of this component is not reflected in performance. ID: In-Domain. OOD: Out-of-Domain.

| Methods | MATH (ID) | | | Coding (OOD) | |
|---|---|---|---|---|---|
| | GSM8K | MATH500 | AMC23 | HumanEval+ | MBPP+ |
| DynaSteer | 82.87 | 74.80 | 47.50 | 67.68 | 54.50 |
| Sentence→Token | 81.27 | 71.80 | 37.50 | 64.63 | 48.94 |
| w/o Pattern Clustering | 81.73 | 72.80 | 45.00 | 66.46 | 53.17 |
| LDA→SVD | 81.88 | 73.20 | 42.50 | 66.46 | 53.17 |
| LDA→MD | 81.50 | 72.60 | 42.50 | 65.85 | 52.91 |
| w/o Entropy Monitoring♣ | 81.73 | 75.20 | 45.00 | 67.07 | 55.56 |
| w/o Temporal Decay♣ | 82.94 | 74.00 | 47.50 | 67.68 | 54.50 |

**Comparison with Recent Literature.** As shown in Table 1, we compare DynaSteer with Plain generation, "Wait" prompt, and various Representation Engineering (RepE) methods (ITI, ACT, and DRESS) across three MATH datasets and two Coding datasets. To comprehensively evaluate the universality of DynaSteer, we employ backbone LLMs with diverse scales, ranging from 1.7B to 8B parameters. The experimental results demonstrate that DynaSteer achieves the best performance in all cases, consistently surpassing strong baselines like ACT and DRESS. Moreover, compared to the second-best performance, DynaSteer yields an average relative improvement of 4.36%↑ on MATH tasks and 3.60%↑ on Coding tasks across all settings, with a maximum improvement of 11.76%↑ on the AMC23 dataset. Notably, the "Wait" prompt method performs comparably to or only marginally better than the Plain baseline in the majority of scenarios. This supports our earlier claim that simple prompting strategies are insufficient to effectively elicit the model's potential capabilities, making it challenging to handle complex reasoning tasks without internal intervention. Furthermore, we provide additional experimental results comparing DynaSteer against TruthX (Zhang et al., 2024) and TruthFlow (Wang et al.) baselines in Appendix F.

**Ablation Studies.** We conduct ablation studies on all eval-

uation datasets using Qwen3-1.7B. As shown in Table 2, substituting our sentence-level truth steering with token-level editing or removing the Pattern Clustering module results in substantial performance degradation. Furthermore, we investigate the choice of truth purification algorithms by replacing Fisher-LDA with the conventional Mean Difference (MD) method and Singular Value Decomposition (SVD). Both alternatives lead to noticeable performance drops. While SVD is numerically stable, it is an unsupervised approach that captures the maximum variance direction, which is not necessarily the most discriminative for separating truth from fallacy. In contrast, Fisher-LDA directly optimizes the between-class over within-class variance ratio using explicit *Truth* labels, extracting a significantly more robust and precise steering direction. These results collectively validate the necessity of the proposed components in enabling DynaSteer to accurately identify and extract truthful representations. We further investigate two additional variants: removing Entropy Monitoring, i.e., applying truth steering to all sentences, and removing Temporal Decay. Notably, eliminating these components does not lead to a significant drop in performance. This is because steering low-entropy sentences is largely ineffective, as the model exhibits high confidence on such sentences and they are therefore difficult to edit meaningfully. Similarly, sentences appearing at the end of the reasoning trajectory often correspond to deeply entrenched errors, making it difficult to fully correct the model's behavior. Although these designs do not provide clear performance gains, we emphasize their advantages in inference efficiency. By editing only a subset of sentences, Entropy Monitoring and Temporal Decay reduce inference token consumption by approximately 56.22%↓ and 52.19%↓, respectively. Detailed experimental results can be found in Appendix. F.6.

# 8. Limitation, Discussion and Conclusion

Despite the promising results obtained in our work, it is important to acknowledge the limitations. Current mainstream research regarding guiding LLMs through reasoning forks predominantly relies on Reinforcement Learning (RL). Notably, Wang et al. (2025a) explicitly demonstrates that leveraging GRPO on reasoning forking tokens can substantially

elevate the model's reasoning capabilities. We compared `DynaSteer` against GRPO under identical experimental settings. The results indicate that GRPO achieves a performance of 84.6% (6.4% ↑), whereas `DynaSteer` achieves 82.3% (4.0% ↑). We hypothesize that this performance gap primarily stems from the geometric nature of the *Truth*. `DynaSteer` assumes that *Truth Steering* can be captured by **linear steering vectors**. However, the actual manifold of *Truth* within complex reasoning trajectories involves **high-dimensional, non-linear dependencies**. RL methods like GRPO update the underlying policy parameters, enabling the model to internalize these complex, non-linear mappings more effectively than an external linear intervention.

Nevertheless, we claim a stark contrast on computational cost: GRPO requires approximately 160GB of VRAM and spans up to 15 hours of training time to converge. In contrast, `DynaSteer` is a training-free framework that directly modulates the LLM during inference. Consequently, while RL methods offer a higher upper bound in performance, `DynaSteer` makes a distinct contribution to resource-constrained communities.

**Conclusion.** In this work, we present a systematic exploration of Representation Edit into the dynamics of *Truth* within LLM reasoning trajectories, moving beyond static attribute editing to dynamic process control. Our investigation establishes the theoretical foundations for dynamic intervention: confirming that the *Truth* exists but is geometrically entangled with reasoning patterns, and revealing that the window for effective correction is transient, constrained by the *Decay Effect*.

To operationalize these insights, we introduce `DynaSteer`, a RepE framework that leverages pattern clustering and Fisher-LDA, `DynaSteer` successfully disentangles the *Truth Steering* from other high-level semantic concepts, addressing the safety risks inherent in naive representation editing. Furthermore, our entropy-aware monitoring ensures that intervention is precise and economical, applied only at critical reasoning forks. Empirical results across diverse benchmarks confirm that `DynaSteer` not only mitigates hallucinations but also achieves superior performance compared to strong baselines. Ultimately, this work offers a resource-efficient alternative to reinforcement learning for steering LLMs, paving the way for more reliable, self-correcting AI systems that seek *Truth* from the very inception of their reasoning.

## Impact Statement

This paper presents work whose goal is to advance the field of machine learning. There are many potential societal consequences of our work, none of which we feel must be specifically highlighted here.

## Acknowledgements

This work was supported by the National Natural Science Foundation of China (62402017), Prevention and Control of Emerging and Major Infectious Diseases-National Science and Technology Major Project (2025ZD01906000, 2025ZD01906004), Beijing Natural Science Foundation (L244063, L244025), and Peking University (Clinical Medicine Plus X Pilot Program-Key Technologies Project 2024YXXLHGG007; "TengYun" Clinical Research Program TY2025015) .

Liantao Ma was supported by the Beijing Traditional Chinese Medicine Science and Technology Development Fund (BJZYZD-2025-13), Young Elite Scientists Sponsorship Program of the Beijing High Innovation Plan (20250628), and Xuzhou Municipal Health Commission Project (2025DF03).

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

# A. Related Work

**Representation Editing for *Truth*** Representation Editing (RepE) has emerged as a powerful paradigm for interpretability and control, aiming to locate and intervene on high-level concepts encoded within the linear activation space of LLMs (Zou et al., 2023; Liao et al., 2026a; Fang et al., 2026b). In the context of *Truth*, Li et al. (2023) proposed Inference-Time Intervention (ITI), which identifies attention heads correlated with truthfulness and shifts their activations to suppress hallucinations. Similarly, Marks & Tegmark (2023) investigated the geometry of truth, revealing that truthful statements often align along specific linear directions in the representation space regardless of the topic. However, a critical limitation of these approaches is their reliance on *static* editing derived from datasets containing explicit answers or simple true/false statements. *There is a notable lack of exploration into dynamic representation editing, specifically regarding how to calculate truth directions within unfolding reasoning trajectories where the final answer is latent and not yet observable.*

**Reinforcing LLM Reasoning** Enhancing the reasoning capabilities of Large Language Models has predominantly relied on Reinforcement Learning (RL) paradigms. Recent advancements have shifted focus from outcome-based supervision to process-oriented approaches, where the model is guided at critical "reasoning forks". For instance, Lightman et al. (2023) demonstrated that Process Reward Models (PRMs) can significantly improve mathematical reasoning by verifying intermediate steps. Furthermore, Shao et al. (2024) introduced Group Relative Policy Optimization (GRPO), which leverages group-level comparisons at reasoning forks to efficiently steer the model toward correct trajectories, yielding substantial performance gains. Wang et al. (2025a) further demonstrates that leveraging GRPO on reasoning forking tokens can substantially elevate the model's reasoning capabilities. *Despite their effectiveness, these RL-based methods suffer from prohibitive computational costs, often requiring massive GPU resources (e.g., hundreds of gigabytes of VRAM) and extensive training time, rendering them largely inaccessible to resource-constrained research communities.*

# B. Derivation of the Fisher-LDA Steering Vector

In this section, we provide the detailed mathematical derivation for the purified steering vector $\mathbf{w}^*$ used in Section 5.

**Rayleigh Quotient Objective** Let $\mathcal{D}^+ = \{\mathbf{h}_i^+\}_{i=1}^{N^+}$ and $\mathcal{D}^- = \{\mathbf{h}_j^-\}_{j=1}^{N^-}$ denote the sets of hidden state activations (defined in Section 2) corresponding to *Truth* and *Fallacy* trajectories, respectively. Let $N = N^+ + N^-$ be the total sample size. Let $\boldsymbol{\delta} = \boldsymbol{\mu}^+ - \boldsymbol{\mu}^-$ be the mean difference vector, where $\boldsymbol{\mu}^+$ and $\boldsymbol{\mu}^-$ are the class centroids. Consistent with Eq. 1 and the definition in Section 5, the within-class covariance matrix $\boldsymbol{\Sigma}_w$ is defined as:

$$\boldsymbol{\Sigma}_w = \sum_{c \in \{+,-\}} \frac{1}{|\mathcal{D}^c|} \sum_{\mathbf{h} \in \mathcal{D}^c} (\mathbf{h} - \boldsymbol{\mu}^c)(\mathbf{h} - \boldsymbol{\mu}^c)^\top. \tag{13}$$

To find the optimal direction $\mathbf{w}$, we maximize the Rayleigh Quotient $\mathcal{J}(\mathbf{w})$, which represents the ratio of the **inter-class signal** to the **intra-class noise**:

$$\mathcal{J}(\mathbf{w}) = \frac{(\mathbf{w}^\top \boldsymbol{\delta})^2}{\mathbf{w}^\top \boldsymbol{\Sigma}_w \mathbf{w}}. \tag{14}$$

By minimizing the denominator, we penalize directions aligned with high-variance outlier dimensions, thereby aligning $\mathbf{w}$ with the most discriminative *Truth* features.

**Analytical Solution and Regularization** To maximize $\mathcal{J}(\mathbf{w})$, we set its gradient with respect to $\mathbf{w}$ to zero. Using the quotient rule:

$$\nabla_{\mathbf{w}} \mathcal{J}(\mathbf{w}) = \frac{2(\mathbf{w}^\top \boldsymbol{\delta}) \boldsymbol{\delta} (\mathbf{w}^\top \boldsymbol{\Sigma}_w \mathbf{w}) - (\mathbf{w}^\top \boldsymbol{\delta})^2 (2 \boldsymbol{\Sigma}_w \mathbf{w})}{(\mathbf{w}^\top \boldsymbol{\Sigma}_w \mathbf{w})^2} = \mathbf{0}. \tag{15}$$

Assuming $\mathbf{w}^\top \boldsymbol{\delta} \neq 0$, we can divide by $2(\mathbf{w}^\top \boldsymbol{\delta})$ and rearrange the terms to obtain the generalized eigenvalue problem:

$$\left( \frac{\mathbf{w}^\top \boldsymbol{\Sigma}_w \mathbf{w}}{\mathbf{w}^\top \boldsymbol{\delta}} \right) \boldsymbol{\delta} = \boldsymbol{\Sigma}_w \mathbf{w}. \tag{16}$$

Since the term in the parenthesis is a scalar $\lambda$, this implies $\boldsymbol{\Sigma}_w \mathbf{w} \propto \boldsymbol{\delta}$. Assuming $\boldsymbol{\Sigma}_w$ is invertible, the analytical closed-form solution is:

$$\mathbf{w}^* \propto \boldsymbol{\Sigma}_w^{-1} \boldsymbol{\delta} = \boldsymbol{\Sigma}_w^{-1} (\boldsymbol{\mu}^+ - \boldsymbol{\mu}^-). \tag{17}$$

In practical LLM scenarios, $\boldsymbol{\Sigma}_w$ may become singular when the number of samples $N < d$. Even when $N > d$, the matrix often becomes ill-conditioned due to high multi-collinearity. To ensure numerical stability, we employ shrinkage estimation to obtain a positive-definite estimate:

$$\hat{\boldsymbol{\Sigma}}_w = (1 - \zeta)\boldsymbol{\Sigma}_w + \zeta\mathbf{I}, \tag{18}$$

where $\zeta \in [0, 1]$ is the shrinkage coefficient (determined by the Ledoit-Wolf estimator). The final discriminant axis used in DynaSteer is $\mathbf{w}^* = \hat{\boldsymbol{\Sigma}}_w^{-1}\boldsymbol{\delta}$.

## C. Notations and Algorithm

To facilitate understanding of the proposed framework, we first summarize the mathematical notations used throughout this paper in Table 3. Subsequently, we present the procedural details of the DynaSteer algorithm.

*Table 3.* Summary of Notations.

| Symbol | Description |
|---|---|
| $\mathcal{M}$ | The autoregressive Large Language Model |
| $L, N, d$ | Number of layers, heads, and hidden dimension |
| $\mathbf{h}_t^{(l,n)}$ | Hidden state at layer $l$, head $n$, token $t$ |
| $\mathcal{T}$ | Reasoning trajectory composed of sentences $(S_1, \ldots, S_T)$ |
| $y \in \{+1, -1\}$ | Binary label for Truth $(+1)$ and Fallacy $(-1)$ |
| $\mathcal{D}$ | Contrastive dataset of hidden states |
| $\mathcal{H}_{\text{top}}$ | Set of identified truth-relevant attention heads |
| $H(S)$ | Shannon entropy of the token distribution for sentence $S$ |
| $\gamma$ | Entropy threshold for detecting reasoning forks |
| $M$ | Number of latent reasoning pattern clusters |
| $\mathcal{C}_m$ | The $m$-th reasoning cluster |
| $\boldsymbol{\Sigma}_{w,m}$ | Within-class covariance matrix for cluster $m$ |
| $\mathbf{v}_m$ | Purified steering vector for cluster $m$ (Fisher-LDA) |
| $\eta_m(\mathbf{h})$ | Dynamic intervention strength for cluster $m$ |
| $\beta(t)$ | Temporal gating coefficient for the Decay Effect |
| $\tilde{\mathbf{h}}$ | The edited hidden state after intervention |

### C.1. Algorithm Description

The DynaSteer framework operates in two distinct phases: an offline preparation phase for steering vector derivation and an online inference phase for dynamic intervention. The complete procedure is outlined in Algorithm 1.

In the **Offline Phase**, we first construct a consistency-based dataset by filtering reasoning trajectories that lead to consensus answers. We then identify the truth-relevant heads ($\mathcal{H}_{\text{top}}$) via linear probing. To handle reasoning heterogeneity, difference vectors are clustered into $M$ patterns. For each cluster, we compute a purified steering vector $\mathbf{v}_m$ using Fisher-LDA to maximize the separation between Truth and Fallacy while minimizing within-class variance.

In the **Online Phase**, the model generates reasoning chains sentence by sentence. We monitor the entropy $H(S_{\text{cand}})$ of each candidate sentence. If the entropy exceeds the threshold $\gamma$ (indicating a reasoning fork) and the step is within the effective temporal window ($\beta(t) = 1$), we trigger the intervention. The current hidden state is projected onto the pre-computed clusters to determine adaptive strengths $\eta_m$, and the aggregated steering is applied. Crucially, a *rollback* mechanism is employed: the unsteered candidate sentence is discarded, and the model regenerates the step using the edited representation $\tilde{\mathbf{h}}$.

## D. Prompts Used in This Work

**Prompt used for solving Math problems**   In our assessment of DynaSteer performance, we employ the following prompts to address relevant Math tasks, including GSM8K (Li et al., 2024), MATH500 (Hendrycks et al., 2021) and AMC23 (Hendrycks et al., 2021) datasets.

---

**Algorithm 1** `DynaSteer`: Dynamic Reasoning Editing

---

**Require:** Model $\mathcal{M}$, Prompt $x$, Thresholds $\gamma, \tau_{\text{decay}}$
**Require:** Pre-computed Clusters $\{\mathcal{C}_m\}_{m=1}^M$, Steering Vectors $\{\mathbf{v}_m\}_{m=1}^M$, Heads $\mathcal{H}_{\text{top}}$
 1: **Initialize:** Trajectory $\mathcal{T} \leftarrow \emptyset$, Step $t \leftarrow 0$
 2: **while** not end of generation **do**
 3:     $t \leftarrow t + 1$
 4:     // Generate candidate sentence without intervention
 5:     $S_{\text{cand}} \leftarrow \mathcal{M}.\text{generate\_sentence}(x, \mathcal{T})$
 6:     Calculate entropy $H(S_{\text{cand}})$
 7:     // Check Spatial (Entropy) and Temporal (Decay) Locus
 8:     **if** $H(S_{\text{cand}}) > \gamma$ **and** $t \leq \tau_{\text{decay}}$ **then**
 9:         $\mathbf{h} \leftarrow \text{Get\_Activations}(\mathcal{M}, x, \mathcal{T})|_{\mathcal{H}_{\text{top}}}$
10:         $\Delta_{total} \leftarrow \mathbf{0}$
11:         **for** $m = 1$ **to** $M$ **do**
12:             Calculate strength $\eta_m(\mathbf{h})$ using Eq. 9
13:             $\Delta_{total} \leftarrow \Delta_{total} + \eta_m(\mathbf{h}) \cdot \mathbf{v}_m$
14:         **end for**
15:         // Apply Aggregated Steering and Rollback
16:         $\tilde{\mathbf{h}} \leftarrow \mathbf{h} + \Delta_{total}$
17:         $\mathcal{M}.\text{set\_activations}(\tilde{\mathbf{h}})$
18:         $S_{\text{steered}} \leftarrow \mathcal{M}.\text{generate\_sentence}(x, \mathcal{T})$
19:         $\mathcal{T} \leftarrow \mathcal{T} \cup \{S_{\text{steered}}\}$
20:     **else**
21:         // Keep original sentence
22:         $\mathcal{T} \leftarrow \mathcal{T} \cup \{S_{\text{cand}}\}$
23:     **end if**
24: **end while**
     Return $\mathcal{T}$

---

**Prompt 1**

INSTRUCTION = r"""
Solve the following math problem step by step. The last line of your response should be of the form Answer: boxedAnswer where Answer is the answer to the problem.
problem
Remember to put your answer on its own line after "Answer:".
"""

**Prompt used for solving Coding problems**   . In our assessment of `DynaSteer` performance, we employ the following prompts to address relevant Coding task on HumanEval+ (Chen et al., 2021) and MBPP+ (Austin et al., 2021) datasets.

**Prompt 2**

MATH_WAIT = r"""
Wait
"""

CODE_WAIT = r"""
Wait, let me rethink about it and give the final Python script.
```python
"""

# E. Experimental Setups

**Network Architecture, Datasets and Metrics**  Our experiments were based on various backbone LLMs, including Qwen3 series (Bai et al., 2023) of various sizes (Qwen3-1.7B and Qwen3-8B) and LLaMA3.2-3B-Instruct (Touvron et al., 2023). We collected *Truth* and *Fallacy* representation using GSM8K (Li et al., 2024) and MATH (Hendrycks et al., 2021) training set to calculate *Truth Steering*. We introduced two types of tasks, **Math** (in-distribution) and **Coding** (out-of-distribution), to assess the effectiveness of `DynaSteer`. For the **Math** tasks, we considered the following datasets: GSM8K test part, MATH500 (Hendrycks et al., 2021) and AMC23 (Hendrycks et al., 2021). For the **Coding** tasks, we considered the HumanEval+ (Chen et al., 2021) and MBPP+ (Austin et al., 2021) datasets. We evaluated these datasets with pass@1 accuracy.

**Baselines**  We adopt the following state-of-the-art approaches as our compared baselines.

- **Plain**: Plain does not use any techniques, LLM directly reason and answer questions.

- **"Wait"**: "Wait" monitors entropy in real-time and manually injects *Wait* prompts at high entropy locations to guide LLM to perform more thinking.

- **RepE literature**: We adopt several recently proposed RepE literature as our baseline models, including ITI (Li et al., 2023), ACT (Wang et al., 2025c) and DRESS (Ma et al., 2025).

The details of the baseline RepE algorithm we use is as follows:

1. **ITI** (Li et al., 2023) (NeurIPS'23): Inference-Time Intervention (ITI) is a lightweight method that improves the truthfulness of large language models by adjusting internal activations along learned directions at inference time, using only a small subset of attention heads. ITI also exposes a controllable trade-off between truthfulness and helpfulness by tuning the intervention strength, suggesting that LLMs encode latent representations of truth internally.

2. **ACT** (Wang et al., 2025c) (WWW'25): Adaptive Activation Steering (ACT) is a tuning-free inference-time method that enhances LLM truthfulness by adaptively steering model activations toward linearly encoded *Truth Steering*. By leveraging multiple truthfulness-related steering vectors and dynamically adjusting intervention strength, ACT effectively mitigates diverse hallucination types. As a plug-and-play add-on, it achieves substantial truthfulness gains across a wide range of models and scales robustly to large LLMs.

3. **DRESS** (Ma et al., 2025) (ICLR'25): DRESS is a lightweight, train-free representation editing method for generating stylized LLM responses while preserving the original semantics. It disentangles a style-relevant subspace in the model's representations and applies adaptively scaled steering vectors to balance stylistic fidelity and semantic integrity.

**Implement Details**  We implemented our framework using PyTorch and the Hugging Face `transformers` library (Wolf, 2019). To enable high-throughput inference, we integrated `vLLM` (Kwon et al., 2023) as the generation backend. For mathematical reasoning benchmarks, we utilized `vLLM` for sampling and employed the `math-verify` library[1] to rigorously parse and validate answer correctness. For code generation tasks, we strictly followed the evaluation protocols and settings established by EvalPlus (Liu et al., 2023). All experiments were conducted on a single NVIDIA A800 (80GB) GPU.

**Construction of Contrastive Pairs.**  To reliably estimate the *Truth Steering* vector, we meticulously construct positive and negative contrastive samples from the training set. Specifically, we prompt the LLM to generate reasoning trajectories. At each generated sentence boundary, we compute the lookahead entropy. When the entropy exceeds a pre-defined threshold—indicating a critical reasoning fork—we pause the primary generation and conduct $N$ independent stochastic rollouts from that exact state (in our experiments, $N = 10$). We then evaluate the final answers of all $N$ rollouts. If all rollouts yield the correct answer, the hidden representations of the current reasoning fork are assigned the label *Truth* ($+1$). Conversely, if all rollouts yield an incorrect answer, they are labeled as *Fallacy* ($-1$). To ensure geometric purity of the representations and to eliminate noise, any ambiguous cases—where the rollouts produce a mix of correct and incorrect answers—are strictly discarded. This highly rigorous filtering strategy guarantees that the resulting contrastive pair dataset robustly captures the intrinsic directional difference between truth and fallacy.

---

[1] https://github.com/huggingface/math-verify

# F. Additional Experiment

## F.1. t-SNE Visualization of Pattern Clustering

To empirically validate the hypothesis of *Reasoning Heterogeneity*, we visualize the geometric distribution of the extracted *Truth Steering* (i.e., the difference vectors $\mathbf{h}^+ - \mathbf{h}^-$) using t-SNE dimensionality reduction. As illustrated in Fig 7, the steering vectors do not coalesce into a single, monolithic distribution; instead, they spontaneously segregate into distinct, well-separated clusters. This visual evidence strongly suggests that the *Truth* signal is not encoded as a universal global direction. Rather, it is realized through diverse geometric orientations that are specific to the underlying reasoning patterns, thereby corroborating the necessity of our pattern clustering approach to capture these local directional nuances.

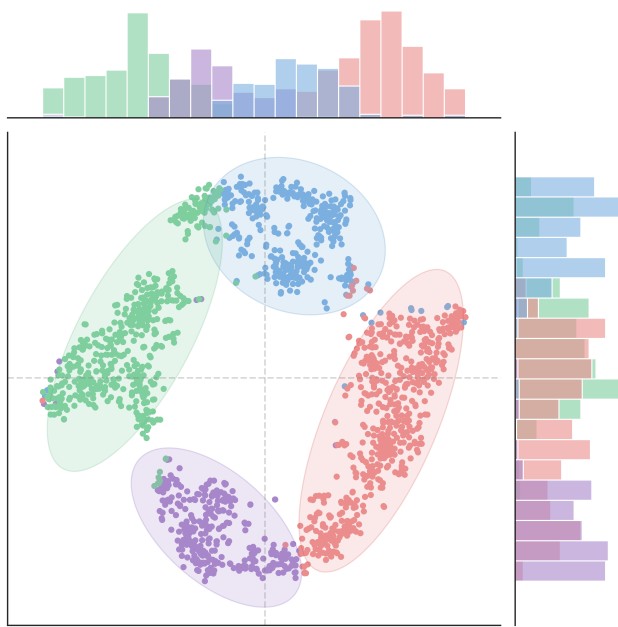

*Figure 7.* t-SNE Visualization of Pattern Clustering.

## F.2. Comparison with Causal Intervention Baselines

To further validate the effectiveness of our probing-based truth localization strategy, we compare `DynaSteer` against two prominent causal intervention methods, ROME (Meng et al., 2022) and MEMIT (Meng et al.). These methods explicitly target specific attention heads based on causal tracing to edit associations within the model. The experiment is conducted using the Qwen3-8B model as the backbone.

*Table 4.* Performance comparison between `DynaSteer` and causal intervention baselines using Qwen3-8B. ID: In-Domain. OOD: Out-of-Domain.

| Methods | MATH (ID) | | | Coding (OOD) | |
|---|---|---|---|---|---|
| | GSM8K | MATH500 | AMC23 | HumanEval+ | MBPP+ |
| ROME | 77.94 | 82.80 | 62.50 | 77.44 | 55.82 |
| MEMIT | 76.95 | 80.40 | 57.50 | 73.78 | 53.70 |
| DynaSteer | **94.39** | **86.20** | **72.50** | **81.71** | **65.61** |

As shown in Table 4, `DynaSteer` substantially outperforms both ROME and MEMIT across all benchmarks. We attribute this performance gap to a fundamental architectural mismatch: causal tracing methods primarily intervene on discrete, individual heads, operating under the assumption that specific concepts or factual associations are highly localized. However, according to the Linear Representation Hypothesis, abstract cognitive signals such as *Truth* are likely encoded as distributed

directions across the residual stream. While an exhaustive combinatorial search for multi-head causal interactions entails exponential complexity and is thus infeasible, our probing-based localization naturally accommodates such distributed and entangled representations, yielding more precise and effective steering.

### F.3. Comparison with TruthX and TruthFlow Baselines

We present additional comparisons with two recent representation editing methods, TruthFlow (Wang et al.) and TruthX (Zhang et al., 2024), focusing on the Qwen3-1.7B backbone.

Table 5. Performance comparison of `DynaSteer` against TruthX and TruthFlow on Qwen3-1.7B.

| Methods | MATH (ID) | | | Coding (OOD) | |
| --- | --- | --- | --- | --- | --- |
| | GSM8K | MATH500 | AMC23 | HumanEval+ | MBPP+ |
| TruthX | 77.71 | 68.00 | 35.00 | 64.63 | 51.06 |
| TruthFlow | 79.23 | 69.00 | 35.00 | 64.02 | 52.12 |
| DynaSteer | **82.87** | **74.80** | **47.50** | **67.68** | **54.50** |

Table 5 illustrates that `DynaSteer` achieves considerable performance gains over both TruthFlow and TruthX. A key architectural distinction lies in how the steering is dynamically adapted. TruthFlow computes a static steering direction once at the beginning of the generation based on the user query. Conversely, `DynaSteer` dynamically monitors and adapts the steering vectors in response to the continuously evolving reasoning state, enabling real-time corrections at critical high-entropy forks. On the other hand, TruthX constructs a global mean-difference vector within a pre-trained truthful subspace, implicitly assuming a uniform truth direction. This assumption is precisely what `DynaSteer` challenges and resolves via pattern clustering, demonstrating that capturing reasoning-specific geometric orientations is vital for optimizing representation editing during complex derivations.

### F.4. Discussion on Attention Probes

In Section 3, we employed minimal-capacity linear probes to ascertain the intrinsic linear separability of *Truth*. Here, we explore the impact of replacing linear probes with more expressive attention-based probes for temporal aggregation. While attention probes elevate the detection accuracy of *Truth* from approximately $78\%$ to $84\%$, we observe that the downstream task performance remains comparable, as shown in Table 6.

Table 6. Ablation study on the choice of probe architecture using Qwen3-1.7B.

| Probe Type | MATH (ID) | | | Coding (OOD) | |
| --- | --- | --- | --- | --- | --- |
| | GSM8K | MATH500 | AMC23 | HumanEval+ | MBPP+ |
| Attention Probe | 82.18 | **75.00** | 47.50 | **68.29** | 53.70 |
| Linear Probe | **82.87** | 74.80 | 47.50 | 67.68 | **54.50** |

This parity in end-to-end performance is expected within our framework. Although attention probes enhance the precision of identifying truth-relevant features during the initial localization phase, the actual steering vectors injected during inference are independently derived by the subsequent Fisher-LDA optimization. Since the efficacy of Fisher-LDA relies predominantly on the geometric distribution of the aggregated hidden representations rather than the capacity of the upstream discriminator, the generative accuracy is not fundamentally constrained by the choice between linear and attention probes.

### F.5. Generalization on Multi-hop QA

To further attest to the generalization capabilities of `DynaSteer` beyond mathematical reasoning and code generation, we extended our evaluation to multi-hop Question Answering (QA) tasks using HotpotQA and 2WikiQA. Multi-hop QA inherently demands dynamic adjustments of reasoning trajectories based on intermediate retrieved knowledge or partial deductions. We utilize Qwen3-8B as the underlying language model for this experiment.

The results in Table 7 illustrate that `DynaSteer` consistently surpasses all benchmark RepE methodologies on both datasets.

*Table 7.* Generalization performance on Multi-hop QA benchmarks using Qwen3-8B.

| Methods | HotpotQA | 2WikiQA |
|---------|----------|---------|
| Base Model | 37.40 | 31.60 |
| ITI | 47.30 | 43.70 |
| ACT | 54.60 | 41.30 |
| DRESS | 50.20 | 48.50 |
| DynaSteer | **55.40** | **52.30** |

This confirms that the pattern-disentangled, dynamically adaptive steering mechanism orchestrated by `DynaSteer` possesses broad applicability. It effectively captures universally applicable cognitive primitives—such as logical bridging, context integration, and fact-checking—that transcend specific task domains, corroborating its robustness and generalizability.

### F.6. Ablation Study on Additional Token Cost

To evaluate the computational overhead of `DynaSteer`, we measure the **Additional Token Cost** incurred during inference. This metric quantifies the extra tokens generated due to the rollback-and-regenerate mechanism triggered by interventions. We compare the full framework against two ablated variants on both In-Domain (MATH) and Out-of-Domain (Coding) tasks using Qwen3-1.7B. The results are presented in Table 8.

**Impact of Entropy Monitoring (Spatial Efficiency)**   The removal of entropy-based filtering (*w/o Entropy Monitoring*) results in a dramatic increase in token cost across all benchmarks. For instance, on the OOD **HumanEval+** dataset, the cost surges from 16.88 to 175.67 ($\sim 10\times$ increase). Similarly, on **MATH500**, the cost more than doubles. This validates our hypothesis regarding the **Spatial Locus** (Sec. 4): the vast majority of generation steps are deterministic syntactic transitions. Without entropy monitoring, the model attempts to intervene at non-critical points, leading to unnecessary rollbacks that do not contribute to reasoning correction but significantly inflate computational cost.

**Impact of Temporal Decay (Temporal Efficiency)**   The exclusion of the temporal gating mechanism (*w/o Temporal Decay*) leads to a substantial efficiency degradation, particularly in complex reasoning tasks. On the challenging **AMC23** benchmark, the token cost spikes from 192.48 to 474.63. This empirical finding reinforces the **Decay Effect** theory (Sec. 4). In long-horizon reasoning, errors become "entrenched" in the later stages. Without the temporal decay constraint, the model futilely attempts to intervene on deep-rooted fallacies. These late-stage interventions rarely succeed in recovering the correct trajectory, resulting in repeated failures and wasted generation cycles. Notably, on Coding tasks, the cost remains identical to the baseline, suggesting that for these specific OOD tasks, effective interventions naturally occur early in the generation, making the late-stage decay gate redundant but harmless.

*Table 8.* Ablation Studies on **Additional** Token Cost due to representation using Qwen3-1.7B as backbone LLM on MATH and Coding tasks. ID: In-Domain. OOD: Out-of-Domain.

| Methods | MATH (ID) | | | Coding (OOD) | |
|---------|-----------|----------|-------|--------------|-------|
| | GSM8K | MATH500 | AMC23 | HumanEval+ | MBPP+ |
| DynaSteer | 28.57 | 107.18 | 192.48 | 16.88 | 13.04 |
| w/o Entropy Monitoring | 39.33 | 232.67 | 238.65 | 175.67 | 131.58 |
| w/o Temporal Decay | 47.35 | 251.93 | 474.63 | 16.88 | 13.04 |

### F.7. Hyper-Parameter Analysis

We further investigate several important hyper-parameters introduced in `DynaSteer` to facilitate future applications, including cluster number $M$ in pattern clustering, Intervention intensity coefficient $\alpha$ in Equ. 9, and the number of selected heads $\mathcal{H}_{\text{top}}$ for intervention, $n\_head$. The experimental results are shown in Fig. 8.

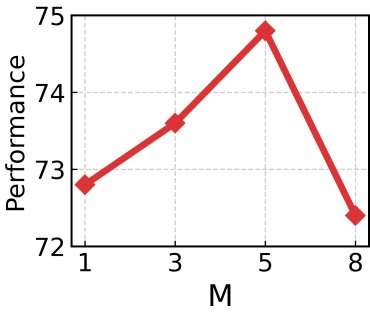 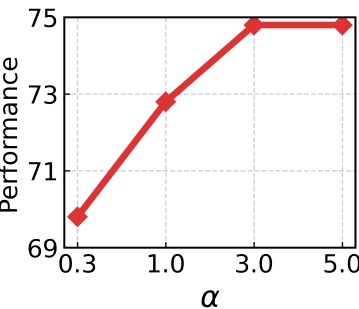 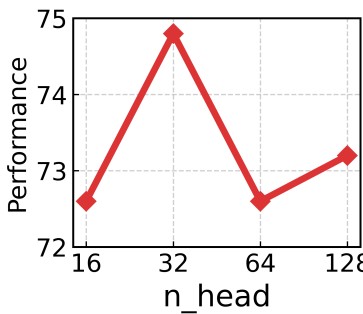

*(a)* Cluster number $M$ in pattern clustering.  *(b)* Intervention intensity coefficient $\alpha$.  *(c)* The number of selected heads $\mathcal{H}_{\text{top}}$ for intervention, $n\_head$.

*Figure 8.* Sensitivity Analysis of Hyper-Parameter.

**Cluster number in pattern clustering, $M$**  We hypothesize that the geometry of truth is entangled with latent reasoning patterns. By varying the cluster number $M \in \{1, 3, 5, 8\}$, we observe the effect of manifold disentanglement.

- **Global vs. Local Geometry:** The baseline $M = 1$ (global mean-difference) yields an accuracy of 72.8%. Increasing $M$ to 5 results in a peak performance of 74.8%, validating our core hypothesis that constructing local linear probes for distinct reasoning patterns captures the truth signal more effectively than a single global direction.

- **Over-Clustering:** However, further increasing $M$ to 8 leads to a performance drop (72.4%). We attribute this to *cluster sparsity*: as the number of clusters increases, the number of samples per cluster decreases, leading to unstable estimation of the covariance matrices for Fisher-LDA.

**Intervention intensity coefficient, $\alpha$**  The coefficient $\alpha$ in Eq. 9 controls the magnitude of the steering vector injection.

- **Steering Efficacy:** As $\alpha$ increases from 0.3 to 3.0, we observe a steady improvement in recovery rates (69.8% $\rightarrow$ 74.8%), indicating that a sufficient magnitude is required to push the model out of the "entrenched" local minima of incorrect reasoning.

- **Safety Saturation:** Notably, performance plateaus at $\alpha = 5.0$ (74.8%) rather than degrading. This contrasts with standard mean-difference steering, where excessive strength often destroys generation coherence. This stability highlights the effectiveness of our *Fisher-LDA purification*: by projecting onto a subspace that minimizes within-class variance, `DynaSteer` ensures that even strong interventions remain orthogonal to essential semantic features, preserving the integrity of the generation.

**The number of selected heads for intervention, $n\_head$**  We analyze the localization of the truth signal by varying the number of manipulated heads.

- **Signal Localization:** The performance exhibits an inverted U-shaped trend, peaking at 32 heads. Using too few heads (16) fails to capture the distributed nature of the truth representation across layers.

- **Noise Introduction:** Conversely, expanding the selection to 64 or 128 heads causes a decline in accuracy (72.6% and 73.2%). This suggests that the truth signal is sparse; including irrelevant heads introduces noise into the aggregated steering vector, diluting the corrective effect. Thus, precise head location is crucial for optimal performance.

# G. Case Studies

To intuitively understand how `DynaSteer` corrects reasoning trajectories, we present a series of qualitative examples from the MATH500 dataset across various mathematical domains. These include both single-step interventions and complex multi-steering scenarios.

## G.1. Single-Steering Case Studies

### G.1.1. TRIGONOMETRIC IDENTITY RETRIEVAL

**Analysis of the Original Trajectory (Fallacy)**   The model's initial inclination (highlighted in **red**) is to employ a **Numerical Approximation** strategy. It attempts to convert the exact symbolic angles $(53°, 81°)$ into floating-point approximations $(\tan 53° \approx 1.3270)$. While mathematically valid in principle, this approach is suboptimal for competitive mathematics. Our post-hoc verification confirmed that trajectories following this "calculation-heavy" pattern consistently failed to reach the exact integer solution.

**Effect of Intervention (Truth)**   Upon detecting high entropy at the reasoning step, `DynaSteer` intervenes. The steered trajectory (highlighted in **green**) activates the model's latent knowledge of high-level trigonometric identities. Specifically, it recalls the conditional identity:

$$\text{If } A + B + C = 180°, \text{ then } \tan A \tan B \tan C = \tan A + \tan B + \tan C.$$

Recognizing this pattern allows the model to bypass calculation entirely and solve the problem via simple linear algebra $(53 + 81 + x = 180)$, instantaneously deriving the correct answer.

---

**Problem Input & Reference**

**Problem:** For some value of $x$, $0 < x < 180$,

$$\tan 53° \tan 81° \tan x° = \tan 53° + \tan 81° + \tan x°.$$

Find $x$.

---

**Reference Answer:** 46

---

**Qwen3-1.7B Reasoning Trajectory**

**Context Generation:**
We are given the equation: $\tan 53° \tan 81° \tan x° = \tan 53° + \tan 81° + \tan x°$
We are to find $x$ such that $0 < x < 180$.

**Intervention Point (Reasoning Fork):**

> *Original Trajectory (Numerical Approximation)*
>
> **Step 1: Use known values**
> We use the known values of the tangents:
>
> - $\tan 53° \approx 1.3270$
>
> - $\tan 81° \approx 6.3138$
>
> We see that both values are positive... *[Leads to approx. error $x \approx 45.8$]*

↓ *DynaSteer Steering*

---

**Steered Trajectory (Symbolic Identity)**

**Step 1: Use known trigonometric identities**
We recognize a useful identity:

$$\tan A \tan B \tan C = \tan A + \tan B + \tan C \quad \text{if } A + B + C = 180°$$

This is a well-known identity in trigonometry.

---

**Continued Generation:**
**Step 2: Match the identity**
In our equation: $\tan 53° \tan 81° \tan x° = \tan 53° + \tan 81° + \tan x°$. This matches the identity above if:

$$53° + 81° + x° = 180°$$

Solve for $x$:

$$x = 180° - 53° - 81° = 46°$$

**Final Answer:** $\boxed{46}$

*Figure 9.* **Case Study on Trigonometry.** The figure contrasts the model's original tendency to perform approximate calculation (Red) with the steered trajectory (Green). DynaSteer successfully guides the model to retrieve the relevant symbolic identity, avoiding calculation errors and directly deriving the integer solution.

### G.1.2. NUMBER BASE CONVERSION

**Analysis of the Original Trajectory (Fallacy)** When converting from binary to octal, the model groups digits into sets of three. However, it fails to pad the binary string to a multiple of three digits beforehand. By grouping directly without padding, the grouping boundaries become misaligned relative to the most significant bits, resulting in an incorrect translation.

**Effect of Intervention (Truth)** DynaSteer intervenes at the critical step of grouping, directing the model to explicitly pad the binary string with leading zeros. This ensures proper 3-bit chunking and yields the correct octal representation.

---

**Problem Input & Reference**

**Problem:** The binary number $10101001110_2$ is equal to what number in base eight?

---

**Reference Answer:** $2516_8$

---

**Qwen3-8B Reasoning Trajectory**

**Context Generation:**
To convert from binary to octal, we group the binary digits into sets of 3, starting from the right. The binary number is $10101001110$.

**Intervention Point (Reasoning Fork):**

**Original Trajectory (Incorrect Grouping)**

**Step 1: Group from the right**
We partition the digits as: $101 \mid 010 \mid 011 \mid 110$.
Converting each group: $5, 2, 3, 6$.
*[Leads to incorrect answer $5236_8$ due to misaligned grouping boundaries]*

---

## ↓ *DynaSteer Steering*

**Steered Trajectory (Correct Grouping with Padding)**

**Step 1: Pad and group correctly**
The number has 11 digits, which is not a multiple of 3. Pad with one leading zero: $\mathbf{0}10 \mid 101 \mid 001 \mid 110$.
Converting each group: $010_2 = 2$, $101_2 = 5$, $001_2 = 1$, $110_2 = 6$.

**Continued Generation:**
**Step 2: Assemble the result**
Concatenating the octal digits in order gives:

$$10101001110_2 = 2516_8$$

**Final Answer:** $\boxed{2516_8}$

---

*Figure 10.* **Case Study on Number Base Conversion.** The model originally misaligns the 3-bit grouping boundaries (Red), producing an incorrect octal representation. `DynaSteer` steers the model to first pad the binary string to a multiple of 3 digits before grouping, yielding the correct conversion.

### G.1.3. LOGARITHMIC EQUATION DOMAIN ANALYSIS

**Analysis of the Original Trajectory (Fallacy)** Faced with finding conditions for "exactly one solution", the model falls back on a common heuristic: setting the discriminant of the resulting quadratic equation to zero. This simplistic approach completely ignores the domain restrictions inherent to logarithmic functions, which can invalidate extraneous roots.

**Effect of Intervention (Truth)** `DynaSteer` steers the model away from the discriminant-only shortcut and prompts a systematic, domain-aware case analysis. By carefully evaluating the roots against the domain constraints, the model successfully identifies the vast majority of valid cases where a positive discriminant yields exactly one valid root.

---

**Problem Input & Reference**

**Problem:** Find the number of integer values of $k$ in the closed interval $[-500, 500]$ for which the equation $\log(kx) = 2\log(x+2)$ has exactly one real solution.

---

**Reference Answer:** 501

---

**Qwen3-8B Reasoning Trajectory**

**Context Generation:**
We rewrite $\log(kx) = \log((x+2)^2)$, so $kx = (x+2)^2$, yielding $x^2 + (4-k)x + 4 = 0$.

**Intervention Point (Reasoning Fork):**

**Original Trajectory (Discriminant-Only Analysis)**

**Step 1: Set discriminant to zero**
For exactly one solution, require $\Delta = (4-k)^2 - 16 = 0$, giving $k = 0$ or $k = 8$. After checking validity, only $k = 8$ works.
*[Concludes only 1 value of $k$, missing the vast majority of valid cases]*

## ↓ *DynaSteer Steering*

---

***Steered Trajectory (Domain-Aware Case Analysis)***

**Step 1: Recognize domain constraints**
The equation requires $kx > 0$ and $x > -2$. Even when the quadratic has *two* real roots, a domain restriction may eliminate exactly one, leaving exactly one valid solution.

---

**Continued Generation:**
**Step 2: Systematic case analysis**
**Case A** ($k > 0$): need $x > 0$ and $x > -2$, so $x > 0$. The quadratic $x^2 + (4 - k)x + 4 = 0$ has product of roots $= 4 > 0$ and sum $= k - 4$. Both roots positive iff $k > 4$, so for $0 < k \le 4$ at most one positive root, and for $k > 8$ ($\Delta > 0$, both roots positive) we get two solutions. For $4 < k < 8$ or $k = 8$, exactly one valid solution.
**Case B** ($k < 0$): need $x < 0$ and $x > -2$, so $-2 < x < 0$. At most one root can fall in this interval.
Careful counting over $k \in [-500, 500]$ yields $\boxed{501}$ valid integer values.
**Final Answer:** $\boxed{501}$

---

*Figure 11.* **Case Study on Logarithmic Equations.** The model originally reduces "exactly one solution" to "discriminant equals zero" (Red), a common reasoning shortcut that ignores how domain restrictions on logarithms can convert a two-root quadratic into a one-solution scenario. DynaSteer steers the model toward a domain-aware case analysis (Green), recovering the full count.

### G.1.4. PIECEWISE FUNCTION SURJECTIVITY

**Analysis of the Original Trajectory (Fallacy)**   To determine surjectivity, the model analyzes the range of the linear piece of the function. It incorrectly assumes the linear piece spans its entire natural range $(-\infty, \infty)$ because the slope is positive, completely overlooking the explicit domain restriction $x < a$.

**Effect of Intervention (Truth)**   DynaSteer forces the model to re-evaluate the range of each piece strictly within its respective domain boundary. This precise range computation leads to the correct no-gap inequality condition ($a^3 \le a^2 + 2a$) necessary for surjectivity.

---

**Problem Input & Reference**

**Problem:** Consider the function $f(x) = \begin{cases} ax^2 & \text{if } x \ge a, \\ ax + 2a & \text{if } x < a, \end{cases}$ where $a$ is some number. What is the largest value of $a$ such that the graph of $y = f(x)$ intersects every horizontal line at least once?

---

**Reference Answer:** 2

---

**Qwen3-8B Reasoning Trajectory**

**Context Generation:**
We need $f$ to be surjective (onto $\mathbb{R}$). The linear piece $ax + 2a$ covers $(-\infty, f(a^-))$ and the quadratic piece $ax^2$ covers $[f(a), +\infty)$ when $a > 0$.

**Intervention Point (Reasoning Fork):**

---

**Original Trajectory (Overlooking Domain Restriction)**

**Step 1: Analyze surjectivity**
The linear piece $ax + 2a$ has slope $a > 0$, so as $x \to -\infty$, $f(x) \to -\infty$. Therefore, the linear part alone covers all horizontal lines.
*[Ignores that the linear piece is only defined for $x < a$, not all of $\mathbb{R}$; concludes $a = 1$ without checking the gap between the two pieces]*

↓ **DynaSteer** *Steering*

**Steered Trajectory (Correct Range Analysis)**

**Step 1: Compute the range of each piece precisely**
For the linear piece ($x < a$): range is $(-\infty, a \cdot a + 2a) = (-\infty, a^2 + 2a)$.
For the quadratic piece ($x \geq a$, $a > 0$): $ax^2$ is increasing on $[a, \infty)$, so the minimum is $a \cdot a^2 = a^3$, giving range $[a^3, \infty)$.

**Continued Generation:**
**Step 2: No-gap condition**
For surjectivity, the ranges must cover all of $\mathbb{R}$, requiring $a^3 \leq a^2 + 2a$:

$$a^3 - a^2 - 2a \leq 0 \implies a(a-2)(a+1) \leq 0$$

For $a > 0$, this gives $0 < a \leq 2$. The largest such $a$ is $\boxed{2}$.
**Final Answer:** $\boxed{2}$

*Figure 12.* **Case Study on Piecewise Functions.** The model originally overlooks the domain restriction of the linear piece ($x < a$) and incorrectly concludes it covers all of $\mathbb{R}$ (Red). DynaSteer steers the model to correctly compute the bounded range of each piece and derive the no-gap inequality $a^3 \leq a^2 + 2a$, identifying $a = 2$ as the critical threshold.

### G.1.5. RECURSIVE SEQUENCE PERIODICITY

**Analysis of the Original Trajectory (Fallacy)**   The model attempts a brute-force computation of the recursive sequence to detect its period. However, it makes a critical arithmetic error at $x_9$. This arithmetic drift cascades into all subsequent terms, permanently derailing the period detection.

**Effect of Intervention (Truth)**   Rather than relying on flawless exhaustive computation, DynaSteer guides the model to recognize the structural anti-symmetry of the recurrence relation ($x_{n+5} = -x_n$). This high-level structural insight immediately establishes a period of 10, enabling a direct and accurate modulo reduction for the final answer.

**Problem Input & Reference**

**Problem:** Given that $x_1 = 211$, $x_2 = 375$, $x_3 = 420$, $x_4 = 523$, and $x_n = x_{n-1} - x_{n-2} + x_{n-3} - x_{n-4}$ for $n \geq 5$, find $x_{531} + x_{753} + x_{975}$.

---

**Reference Answer:** 898

**Qwen3-8B Reasoning Trajectory**

**Context Generation:**
We compute terms to detect periodicity. $x_5 = 523 - 420 + 375 - 211 = 267$, $x_6 = 267 - 523 + 420 - 375 = -211$.

**Intervention Point (Reasoning Fork):**

---

*Original Trajectory (Arithmetic Drift)*

**Step 1: Continue computing**
$x_7 = -211 - 267 + 523 - 420 = -375$, $x_8 = -375 + 211 + 267 - 523 = \mathbf{-420}$
$x_9 = -420 + 375 - 211 + 267 = \mathbf{11}...$
*[Accumulates sign errors at $x_9$ (correct value is $-523$), derailing all subsequent terms and the period detection]*

---

↓ **DynaSteer** *Steering*

---

*Steered Trajectory (Structured Period Analysis)*

**Step 1: Identify the anti-symmetry**
Computing carefully: $x_6 = -211 = -x_1$, $x_7 = -375 = -x_2$, ..., $x_{10} = -267 = -x_5$. The sequence satisfies $x_{n+5} = -x_n$, hence it is **periodic with period 10**.

---

**Continued Generation:**
**Step 2: Reduce indices modulo 10**
$531 \bmod 10 = 1$, $753 \bmod 10 = 3$, $975 \bmod 10 = 5$.
Therefore: $x_{531} + x_{753} + x_{975} = x_1 + x_3 + x_5 = 211 + 420 + 267 = \boxed{898}$.
**Final Answer:** $\boxed{898}$

*Figure 13.* **Case Study on Recursive Sequences.** The model attempts brute-force computation but accumulates arithmetic errors at $x_9$ (Red), derailing period detection. DynaSteer steers the model to identify the anti-symmetry $x_{n+5} = -x_n$ and period-10 structure, enabling direct index reduction and an exact answer.

### G.1.6. COMBINATORIAL PROBABILITY

**Analysis of the Original Trajectory (Fallacy)** In calculating the probability of restoring the original state of the bags, the model recognizes that Bob must return the exact color Alice transferred. However, it overlooks the fact that Bob's bag now contains *two* identical balls of that color. It incorrectly assigns a return probability of $1/6$ instead of $2/6$.

**Effect of Intervention (Truth)** DynaSteer intervenes to structure the conditional probability calculation. By explicitly conditioning on Alice's choice, the model correctly tracks the updated composition of Bob's bag, recognizing the duplicate ball and arriving at the correct $2/6$ probability for the return transfer.

---

**Problem Input & Reference**

**Problem:** Bob and Alice each have a bag that contains one ball of each of the colors: blue, green, orange, red, and violet. Alice randomly selects one ball from her bag and puts it into Bob's bag. Bob then randomly selects one ball from his bag and puts it into Alice's bag. What is the probability that after this process, the contents of the two bags are the same as they were before?

---

**Reference Answer:** $\frac{1}{3}$

---

**Qwen3-8B Reasoning Trajectory**

**Context Generation:**
Alice transfers one of 5 balls to Bob (now 6 balls). Bob then transfers one of 6 balls back to Alice. We need both

bags to return to their original state.

**Intervention Point (Reasoning Fork):**

> ### Original Trajectory (Missing Duplicate Ball)
>
> **Step 1: Compute return probability**
> Bob must return the same color Alice gave. Bob has 6 balls, so $P = \frac{1}{6}$.
> *[Overlooks that Bob now has **two** balls of that color (his original + Alice's), so the correct probability is $\frac{2}{6}$]*



↓ `DynaSteer` *Steering*



> ### Steered Trajectory (Conditional Case Analysis)
>
> **Step 1: Condition on Alice's choice**
> Let Alice transfer color $c$. Bob's bag now has 6 balls: 5 original + one extra $c$.
> For bags to restore, Bob **must return** color $c$. His bag contains **two** copies of $c$ (his original + Alice's), so
> $P(\text{Bob returns } c) = \frac{2}{6} = \frac{1}{3}$.

**Continued Generation:**
**Step 2: Combine**
Since this probability $\frac{1}{3}$ is the same regardless of which color $c$ Alice chose, the total probability is:

$$P = \sum_c P(\text{Alice picks } c) \cdot P(\text{Bob returns } c) = 5 \cdot \frac{1}{5} \cdot \frac{2}{6} = \frac{1}{3}$$

**Final Answer:** $\boxed{\dfrac{1}{3}}$

*Figure 14.* **Case Study on Combinatorial Probability.** The model correctly identifies that Bob must return the transferred color, but overlooks the duplicate ball in Bob's bag, computing $\frac{1}{6}$ instead of $\frac{2}{6}$ (Red). `DynaSteer` steers the model to recognize the duplicate and arrive at the correct probability $\frac{1}{3}$.

## G.2. Multi-Steering Case Studies

In more complex problems, a single intervention may not be sufficient to fully correct a trajectory, as the model may encounter multiple independent or cascading reasoning forks. We demonstrate `DynaSteer`'s capability to apply successive interventions in these multi-steering case studies.

### G.2.1. POLYNOMIAL FACTORIZATION

**Analysis of the Original Trajectory (Fallacy)** The model commits two independent cascading errors. First, during polynomial multiplication, it only considers the single highest-order product for each degree, systematically ignoring all cross-terms. Second, when solving for the coefficients, it treats them as free variables and assigns arbitrary values instead of solving the resulting fully determined linear system.

**Effect of Intervention (Truth)** `DynaSteer` intervenes twice. The first steering corrects the polynomial expansion, ensuring all cross-terms are explicitly enumerated and matched. The second steering prevents the model from guessing arbitrary values, forcing it to properly set up and solve the $4 \times 4$ linear system to find the unique coefficients.

## Problem Input & Reference

**Problem:** The polynomial $x^3 - 3x^2 + 4x - 1$ is a factor of $x^9 + px^6 + qx^3 + r$. Enter the ordered triple $(p, q, r)$.

**Reference Answer:** $(6, 31, -1)$

## Qwen3-8B Reasoning Trajectory (Multi-Steering)

**Context Generation:**
Let $f(x) = x^3 - 3x^2 + 4x - 1$ and $g(x) = x^9 + px^6 + qx^3 + r$. Since $f(x) \mid g(x)$, we write $g(x) = f(x) \cdot h(x)$ where $h(x) = x^6 + a_5x^5 + a_4x^4 + a_3x^3 + a_2x^2 + a_1x + a_0$.

**Intervention Point 1 (Coefficient Matching):**

> **Original Trajectory #1: Incorrect Cross-Term Enumeration**
>
> **Step 3:** Compute the coefficient of $x^8$ in $f(x) \cdot h(x)$:
> "Only $x^3 \cdot a_5x^5 = a_5x^8$, so $a_5 = 0$." Similarly concludes $a_4 = 0$, $a_2 = 0$, $a_1 = 0$.
> *[Error: Considers only the single highest-order product for each degree, systematically ignoring cross-terms such as $(-3x^2)(a_5x^5)$ in $x^7$, $(4x)(a_4x^4)$ in $x^5$, etc. This cascading omission forces all intermediate coefficients to zero.]*

### ↓ DynaSteer *Steering #1*

> **Steered Trajectory #1: Complete Cross-Term Expansion**
>
> **Step 3 (corrected):** The coefficient of $x^8$ involves **all** pairs ($x^i$ from $f$) × ($x^j$ from $h$) with $i + j = 8$:
> $\text{coeff}(x^8) = 1 \cdot a_5 + (-3) \cdot 1 = a_5 - 3 = 0 \Rightarrow a_5 = 3$.
> Similarly: $\text{coeff}(x^7) = a_4 - 3a_5 + 4 = a_4 - 5 = 0 \Rightarrow a_4 = 5$.

> **Continued Generation (after Steering #1):**
> From zero-coefficient constraints on $x^5$ and $x^4$:
> $a_2 - 3a_3 + 4a_4 - a_5 = 0$ and $a_1 - 3a_2 + 4a_3 - a_4 = 0$.
> From $x^1$ and $x^2$: $4a_0 - a_1 = 0$ and $-3a_0 + 4a_1 - a_2 = 0$.

**Intervention Point 2 (System Solving):**

> **Original Trajectory #2: Arbitrary Free-Variable Assignment**
>
> **Step 5:** "We can choose $a_0 = 1$, then $q = 1$, $r = -1$, and pick $p = 0$."
> *[Error: Treats $a_3$ and $a_0$ as free variables due to the earlier zero-coefficient error, then assigns arbitrary values instead of solving the fully determined 4×4 linear system.]*

### ↓ DynaSteer *Steering #2*

> **Steered Trajectory #2: Solve the Determined System**
>
> **Step 5 (corrected):** The system has 4 equations in 4 unknowns $(a_3, a_2, a_1, a_0)$:
> Substituting $a_5=3$, $a_4=5$ into the constraints and solving: $a_3 = 10$, $a_2 = 13$, $a_1 = 4$, $a_0 = 1$.

---

**Final Computation:**
$p = a_3 - 3a_4 + 4a_5 - 1 = 10 - 15 + 12 - 1 = 6, \quad q = a_0 - 3a_1 + 4a_2 - a_3 = 1 - 12 + 52 - 10 = 31,$
$r = -a_0 = -1.$
**Final Answer:** $\boxed{(6, 31, -1)}$

---

*Figure 15.* **Multi-Steering Case Study: Polynomial Factorization.** The model commits two *cascading* errors: (1) it enumerates only single-product terms for each degree, ignoring cross-terms from the three non-leading coefficients of $f(x)$ (Red #1), which collapses the quotient $h(x)$ to $x^6 + a_3 x^3 + a_0$; (2) believing $a_3$ and $a_0$ are free, it assigns arbitrary values instead of solving the determined linear system (Red #2). DynaSteer applies two successive steering interventions at tokens 127 and 246, correcting the cross-term expansion and then the system solving, recovering the correct $(p, q, r) = (6, 31, -1)$.

### G.2.2. GCD/LCM COMBINATORIAL COUNTING

**Analysis of the Original Trajectory (Fallacy)**   The model makes two compounding errors in combinatorial counting. First, it conflates the formulas for ordered and unordered coprime factor pairs, applying $2^{k-1}$ while inconsistently describing it. Second, believing it needs to account for symmetry again, it redundantly divides the already-correct count by 2, halving the final answer.

**Effect of Intervention (Truth)**   DynaSteer applies two successive steering interventions. The first establishes an explicit $2^4 \rightarrow 16$ ordered pair derivation before halving to 8, clarifying the logic. The second intervention triggers at the final count to prevent the spurious redundant division, confidently halting the process at the correct answer of 8.

---

**Problem Input & Reference**

**Problem:** If $a$ and $b$ are positive integers such that $\gcd(a, b) = 210$, $\text{lcm}[a, b] = 210^3$, and $a < b$, how many possible values are there for $a$?

---

**Reference Answer:** 8

---

**Qwen3-8B Reasoning Trajectory (Multi-Steering)**

**Context Generation:**
Write $a = 210x$, $b = 210y$ with $\gcd(x, y) = 1$. Then $xy = 210^2 = 2^2 \cdot 3^2 \cdot 5^2 \cdot 7^2$. We seek coprime pairs $(x, y)$ with $x < y$.

**Intervention Point 1 (Pair-Counting Formula):**

**Original Trajectory #1: Wrong Formula Application**

**Step 5:** "The number of unordered coprime factor pairs is $2^{k-1}$ where $k$ is the number of distinct primes. With $k = 4$: $2^{4-1} = 8$ unordered pairs."
*[Error: Conflates the formula. The correct reasoning is: $2^k = 16$ **ordered** pairs (each of $k=4$ primes goes entirely to $x$ or $y$), and $2^{k-1} = 8$ **unordered** pairs. The model states $2^{k-1} = 8$ as "unordered pairs," which is actually correct at this step, but then proceeds to divide again.]*

↓ *DynaSteer Steering #1*

---

**Steered Trajectory #1: Explicit Pair Enumeration**

**Step 5 (corrected):** Each of the 4 prime-power factors $\{2^2, 3^2, 5^2, 7^2\}$ must go **entirely** to $x$ or $y$ (to maintain coprimality). This gives $2^4 = 16$ ordered pairs.
Since no pair has $x = y$ (that would require $\gcd(x,y) = x \neq 1$), exactly half satisfy $x < y$: **8 unordered pairs**.

---

**Continued Generation (after Steering #1):**
The 8 pairs $(x, y)$ with $x < y$ are: $(1, 44100), (4, 11025), (9, 4900), (25, 1764), (36, 1225), (49, 900), (100, 441), (225, 196)$.

**Intervention Point 2 (Final Count):**

**Original Trajectory #2: Redundant Division by 2**

**Step 7:** "So there are 8 unordered coprime factor pairs. But we want ordered pairs with $x < y$, so we divide by 2: $\frac{8}{2} = 4$."
*[Error: The 8 pairs are **already** unordered with $x < y$; dividing again halves the correct count.]*

$$\downarrow \texttt{DynaSteer} \text{ Steering \#2}$$

**Steered Trajectory #2: Recognize Counting Is Complete**

**Step 7 (corrected):** The 8 pairs listed above already satisfy $x < y$. No further division is needed. Each gives a distinct value $a = 210x$.

**Final Answer:** The number of possible values for $a$ is $\boxed{8}$.

*Figure 16.* **Multi-Steering Case Study: GCD/LCM Counting.** The model makes two compounding errors in combinatorial counting: (1) it applies the formula $2^{k-1}$ without clarifying whether it yields ordered or unordered pairs (Red #1); (2) it then redundantly divides by 2 a second time, halving the already-correct count from 8 to 4 (Red #2). DynaSteer steers at both junctures—first to establish the explicit $2^4 \rightarrow 8$ derivation, then to prevent the spurious halving—recovering the correct answer of 8.

### G.2.3. TRIANGLE TRIGONOMETRY

**Analysis of the Original Trajectory (Fallacy)**   The model falls into a circular reasoning loop. It initially computes $\tan A$ as a symbolic expression involving side lengths $a, b, c$, failing to realize the area constraint implies a unique numeric value. When attempting to solve the trigonometric equation, it re-substitutes the side-length expressions back in, resulting in a useless tautology ($\tan A = \tan A$).

**Effect of Intervention (Truth)**   DynaSteer steers the model out of the loop at two critical junctures. First, it guides the model to substitute the law of cosines into the area formula, successfully eliminating the side lengths and yielding a single-variable equation $\sin A = 4(1 - \cos A)$. Second, instead of re-substituting, it prompts the use of half-angle identities to analytically solve the pure trigonometric equation, yielding the final numeric answer.

---

### Problem Input & Reference

**Problem:** The area of triangle $ABC$ is equal to $a^2 - (b-c)^2$, where $a, b, c$ are the sides of triangle $ABC$, as usual. Compute $\tan A$.

---

**Reference Answer:** $\dfrac{8}{15}$

---

### Qwen3-8B Reasoning Trajectory (Multi-Steering)

**Context Generation:**
From the area formula: $\frac{1}{2}bc\sin A = a^2 - (b-c)^2 = a^2 - b^2 + 2bc - c^2$.
By the law of cosines: $a^2 = b^2 + c^2 - 2bc\cos A$, so $a^2 - b^2 - c^2 = -2bc\cos A$.

**Intervention Point 1 (Substitution Strategy):**

---

*Original Trajectory #1: Direct Ratio Approach (Circular)*

**Step 4:** Compute $\tan A = \frac{\sin A}{\cos A}$ by dividing the area expression by the cosine expression:
$\tan A = \frac{4(a^2 - (b-c)^2)}{b^2 + c^2 - a^2}$.
*[Error: This expression still contains $a, b, c$. The model fails to recognize that the area condition provides an additional constraint that uniquely determines $\tan A$ as a **numeric** constant, independent of specific side lengths.]*

---

## ↓ `DynaSteer` Steering #1

---

*Steered Trajectory #1: Substitute Law of Cosines into Area*

**Step 4 (corrected):** Substitute $a^2 - b^2 - c^2 = -2bc\cos A$ into the area equation:
$\frac{1}{2}bc\sin A = -2bc\cos A + 2bc = 2bc(1 - \cos A)$.
Dividing both sides by $bc$: $\quad \frac{1}{2}\sin A = 2(1 - \cos A)$.

---

**Continued Generation (after Steering #1):**
We now have: $\sin A = 4(1 - \cos A)$. This is a **single-variable** trigonometric equation in $A$.

**Intervention Point 2 (Solving the Trigonometric Equation):**

---

*Original Trajectory #2: Circular Re-substitution*

**Step 5–6:** Attempts to solve $\sin A = 4(1 - \cos A)$ by substituting $\sin A = \frac{2(a^2 - (b-c)^2)}{bc}$ back into the equation, re-introducing $a, b, c$ and arriving at the tautology $\tan A = \tan A$.
*[Error: Instead of using a half-angle identity to solve the pure trigonometric equation, the model loops back to the original side-length expressions.]*

---

## ↓ `DynaSteer` Steering #2

---

---

*Steered Trajectory #2: Apply Half-Angle Identities*

**Step 5 (corrected):** Use half-angle substitutions: $\sin A = 2 \sin \frac{A}{2} \cos \frac{A}{2}$ and $1 - \cos A = 2 \sin^2 \frac{A}{2}$:
$2 \sin \frac{A}{2} \cos \frac{A}{2} = 4 \cdot 2 \sin^2 \frac{A}{2}$.
Since $\sin \frac{A}{2} \neq 0$ for a valid triangle, divide: $\quad \cos \frac{A}{2} = 4 \sin \frac{A}{2}$, i.e., $\tan \frac{A}{2} = \frac{1}{4}$.

---

**Final Computation:**

By the double-angle formula: $\quad \tan A = \dfrac{2 \tan \frac{A}{2}}{1 - \tan^2 \frac{A}{2}} = \dfrac{2 \cdot \frac{1}{4}}{1 - \frac{1}{16}} = \dfrac{\frac{1}{2}}{\frac{15}{16}} = \dfrac{8}{15}$.

**Final Answer:** $\boxed{\dfrac{8}{15}}$

*Figure 17.* **Multi-Steering Case Study: Triangle Trigonometry.** The model encounters two distinct reasoning failures: (1) it computes $\tan A$ as a symbolic expression in $a, b, c$ without recognizing that the area constraint determines a unique numeric value (Red #1); (2) when guided to the single-variable equation $\sin A = 4(1 - \cos A)$, it re-introduces side-length variables instead of applying half-angle identities, producing a tautology (Red #2). DynaSteer steers twice—first to eliminate side lengths via the law of cosines substitution, then to invoke half-angle identities—breaking the circular reasoning and extracting $\tan A = \frac{8}{15}$.

