# OpenReview forum: "Search for Truth from Reasoning: A Dynamic Representation Editing Framework for Steering LLM Trajectories"
_ICML.cc/2026/Conference — ICML 2026 regular_

### Official Review · Reviewer_DXZL · 2026-03-06

**Soundness:** 3
**Presentation:** 2
**Significance:** 3
**Originality:** 3
**Overall Recommendation:** 4
**Confidence:** 5

**Summary:**

This paper explores Representation Editing in dynamic reasoning trajectories, uncovering some critical insights. Specifically, by investigating the geometry of the representation spaces, the authors propose DynaSteer, a dynamic steering framework to improve reasoning accuracy by leveraging clustered steering purified by Fisher-LDA. The experimental results demonstrate that DynaSteer outperforms existing static steering baselines on math and coding benchmarks.

**Compliance With Llm Reviewing Policy:**

Affirmed.

**Final Justification:**

The rebuttal addressed my main concerns. The method itself is validated both from motivation and empirically. I would give a positive assessment if the final manuscript is revised according to the discussion.

**Key Questions For Authors:**

1. As I mentioned in the "Weakness" part, the **purification is not robust**. A similar work, TruthFlow [1], explores SVD as the purification method and achieves good performance on TruthfulQA. I would appreciate it if the authors could provide either a theoretical analysis to compare Fisher-LDA with SVD (especially on robustness or worst case) or an empirical comparison to validate their choice of the purification method. Besides, since the core motivation is quite similar: both work challenges the uniformity of traditional steering vector, i.e., truth may not follow a universal direction in the representation space, I think TruthFlow can be a good baseline to compare, validating the effectiveness of proposed DynaSteer.
2. **The piece-wise linearity assumption:** It motivates the use of k-means to separate the representation space. However, k-means is not guaranteed to separate reasoning patterns within a highly anisotropic representation space. Furthermore, the Fisher-LDA relies heavily on the intra-cluster covariance matrix. If the k-means fails to really find the piece-wise linear truth representation space, the Fisher-LDA will consequently fail to purify the steering vector as desired. In comparison, TruthX [2] "trains" a truth-related subspace where they apply the steering. This is more guaranteed than using clustering, hoping to find an accurate division. Therefore, I'm curious about how to solidify the usage of k-means. And I also expect to see the comparison with TruthX to verify the effectiveness and stability, if possible.
3. **Transferability:** While the empirical improvements on OOD data are appreciated, the mechanism behind this transferability fundamentally contradicts the proposed geometric assumption. According to Sec 6.3, the k-means is designed to disentangle reasoning patterns. If the clusters represent math reasoning patterns, it will be counterintuitive to imagine the local purified steering vectors can seamlessly improve coding (e.g., handle loops, recursion, syntax errors, etc).

[1] Wang, H., Cao, B., Cao, Y., & Chen, J. (2025). Truthflow: Truthful llm generation via representation flow correction. arXiv preprint arXiv:2502.04556.

[2] Zhang, S., Yu, T., & Feng, Y. (2024, August). Truthx: Alleviating hallucinations by editing large language models in truthful space. In Proceedings of the 62nd Annual Meeting of the Association for Computational Linguistics (Volume 1: Long Papers) (pp. 8908-8949).

**Limitations:**

yes

**Strengths And Weaknesses:**

## Strengths
1. The paper investigates an underexplored point, the working mechanism of representation steering in reasoning patterns.
2. The comprehensive explorations motivate the methodology, validating the logic and providing a solid foundation for the proposed DynaSteer framework.
3. Experimental results show SOTA performance across various benchmarks compared to different baselines.

## Weaknesses
1. Presentation: Although the general logic and structure are good, the wording is somewhat misleading or even overly conceptual, making it hard to understand. For example, the authors repeatedly mention the word "Truth", which typically refers to factual accuracy or the mitigation of hallucinations. But it's not until section 3 that the definition of "Truth" is given as reasoning accuracy.
2. Unclear notation and definition: in Section 2, the authors define $h^{(l,n)}$ as the internal activation of the input **sequence**. However, it's necessary to clarify how the hidden state is extracted: the last token? Or the average hidden states over all tokens? Furthermore, in Section 3, the authors compare "token-level" and "sentence-level" without explaining how they extract the hidden states.
3. Robustness of Fisher-LDA: as the authors mention in Appendix B, the purified steering vector requires inverting the covariance matrix $\Sigma_{\omega}^{-1}$, which is frequently ill-conditioned in high-dimensional LLM activation spaces. When $\zeta \rightarrow 1$, the purified steering vector degenerates to mean-difference. Also, the Rayleigh Quotient is sensitive to outliers as it depends on second-order moments. If the activations are heavy-tailed, the covariance matrix may be easily distorted.

---

> ### Author Rebuttal · Authors · 2026-03-31
>
> We sincerely thank the reviewer for the thorough feedback and the recognition of DynaSteer's novelty in mechanistic steering, the well-motivated explorations, and the SOTA performance. Below we address each concern.
>
> **W1: Presentation.** We have revised the manuscript to reduce conceptual abstraction and moved the formal definition of "Truth" (reasoning trajectory correctness) to the Introduction. We apologize for the confusion.
>
> **W2: Hidden State Extraction.** In both token-level and sentence-level settings, the hidden state is the **last token's activation** of the input sequence. The difference: at token-level, the sequence can be truncated at any position; at sentence-level, truncation is restricted to sentence boundaries (punctuation-defined). Sentence-level extraction thus acts as a **filtering mechanism**, ensuring hidden states are extracted only where complete semantic units form — where Truth signals concentrate. Section 3 has been revised accordingly.
>
> **W3: Fisher-LDA Robustness.** The regularization $\zeta$ is **not manually tuned** but computed by Ledoit-Wolf shrinkage (Line 609), providing data-driven optimal shrinkage. Over **ten** random splits, **$\zeta$ stably falls in [0.48, 0.78], well below $\zeta=1$**, confirming that the Fisher-LDA direction incorporates meaningful within-class covariance beyond mean-difference. DynaSteer's sentence-level probing greatly filters out outliers, its clustering further improves $\Sigma_w$ conditioning by ensuring within-cluster homogeneity, greatly reduces the occurrence of outliers within the same cluster.
>
> **Q1: Fisher-LDA vs. SVD, TruthFlow, and TruthX.** We acknowledge that SVD is numerically more stable, but the maximum variance direction captured by SVD may not necessarily be the most discriminative of Truth. Fisher-LDA directly optimizes the between/within-class variance ratio using labels of Truth. Our baseline DRESS [2] already uses SVD and is outperformed by DynaSteer (Table 1). We conducted further ablations and added TruthFlow/TruthX as baselines:
>
> |  | GSM8K | MATH500 | AMC23 | HumanEval+ | MBPP+ |
> |---|:---:|:---:|:---:|:---:|:---:|
> | DynaSteer | **82.87** | **74.80** | **47.50** | **67.68** | **54.50** |
> | Fisher-LDA→SVD | 81.88 | 73.20 | 42.50 | 66.46 | 53.17 |
> | TruthFlow [1] | 79.23 | 69.00 | 35.00 | 64.02 | 52.12 |
> | TruthX [3] | 77.71 | 68.00 | 35.00 | 64.63 | 51.06 |
>
> Key architectural differences: TruthFlow selects a steering direction **once** at generation onset based on the query, while DynaSteer **dynamically adjusts** based on the evolving reasoning state — we believe reasoning correctness requires real-time detection, not one-shot prediction. TruthX computes a **global mean-difference** in its **learned** truthful space, implicitly assuming a uniform truth direction — the very assumption DynaSteer challenges via per-cluster vectors.
>
> **Q2: K-Means Validity and TruthX.** Our key observation is that different reasoning states require **qualitatively different** steering types (e.g., redirecting toward reflection, conceptual deepening, expression refinement). **The exact taxonomy is unknown a priori**. K-means serves as a **lightweight, assumption-minimal** tool to discover these patterns, with $k$ controlling granularity. We validate quality via t-SNE visualization (Figure 7), showing well-separated clusters, and our ablation (Table 3) confirms k-means + Fisher-LDA effectively recovers distinct truth directions. We further provided **detailed cases** for each class cluster in the **anonymous repository** to better introduce our design intentions on k-means to readers.
>
> TruthX's subspace decomposition requires **pre-defining subspace semantics**, feasible for binary truthful/untruthful but **difficult for diverse, unknown reasoning steering types**. We trained a unified truthful space as TruthX's baseline — results above show DynaSteer's consistent advantage.
>
> **Q3: Transferability.** The clusters do **not** represent task-specific patterns (e.g., "algebra") but **general cognitive operations** — steering toward self-verification, conceptual deepening, or summarization. These operations are task-agnostic: a model benefits from verification steering whether solving math or debugging code. We examined cluster assignments on coding tasks and found they map to the same cognitive categories identified on math, confirming that clusters capture **transferable reasoning primitives** rather than domain-specific patterns.
>
> **References**
>
> [1] Truthflow: Truthful llm generation via representation flow correction, ICML'25
>
> [2] Truthx: Alleviating hallucinations by editing large language models in truthful space. ACL'24
>
> [3] Dressing Up LLM: Efficient Stylized Question-Answering via Style Subspace Editing. ICLR'25.

---

> > ### Author Rebuttal · Reviewer_DXZL · 2026-04-03
> >
> > Thanks for the detailed rebuttal, which fully addressed my concerns. Therefore, I'd like to raise the score to 4. I encourage the authors to carefully incorporate what has been discussed into the final manuscript, with a particular focus on improving the presentation.

---

> > > ### Author Response · Authors · 2026-04-03
> > >
> > > Reviewer DXZL,
> > >
> > > Thank you for carefully reading our previous response and for providing such **positive and encouraging feedback**. Your reply is **truly motivating** to us, as it indicates that several issues in the original manuscript have been properly clarified.
> > >
> > > We **sincerely appreciate** your additional suggestions on our manuscript. We are committed to incorporating **all** the content from the rebuttal, including the supplementary materials provided in the anonymous repository, **into our revised manuscript**. We are also grateful for your comments regarding the presentation. We **have invited all co-authors** to place greater emphasis on this aspect and to collaboratively conduct **cross-revisions** to ensure the **clarity** and **accuracy** of the manuscript.
> > >
> > > Please rest assured that we take **all potential issues** in the manuscript very **seriously** and are **fully committed to** addressing them. Every question, doubt, and suggestion you have raised is of immense value to us. We feel **truly fortunate** to have received your **valuable and patient guidance** during the rebuttal period. We also sincerely hope for your continued **guidance** and **supervision** to further advance our work.
> > >
> > > Once again, thank you for your time, your effort, and your thoughtful and patient guidance.
> > >
> > > Best wishes,
> > >
> > > All authors of submission 25560

---

### Official Review · Reviewer_4VJY · 2026-03-12

**Soundness:** 4
**Presentation:** 4
**Significance:** 3
**Originality:** 3
**Overall Recommendation:** 4
**Confidence:** 4

**Summary:**

This paper introduces a steering mechanism of LLM reasoning trajectories towards the true direction. The proposed methodology DynaSteer, identifies the spatio-temporal location of steering based on entropy of reasoning forks and the likelihood of correctness decay over time. Then it learns a true direction from a contrastive sentence level representation difference of reasoning forks and utilize this true direction to steer the model at the determined reasoning fork at a particular sequence step. Experiments on three models and three datasets verify the effectiveness of dynamics steering of reasoning steps in LLMs.

**Compliance With Llm Reviewing Policy:**

Affirmed.

**Final Justification:**

The authors have addressed my concerns regarding the soundness of the results, and I will increase my score for this accordingly.

**Key Questions For Authors:**

Please see the weaknesses.

**Limitations:**

Yes

**Strengths And Weaknesses:**

Strengths:
- The proposed methodology of dynamically steering the reasoning chain of LLM is novel. In particular, the identification of the reasoning fork to be edited based on sentence-level entropy, and the recovery of performance decay over time to determine early vs. late steering effectiveness are very intuitive.
- The paper is well written, and the three research questions are well motivated.
- The ablation study clearly demonstrates the entanglement between the true space and the reasoning space, and the truth space can be further purified by the Fischer-LDA projection.

Weaknesses:
- Methodological details are missing:
  - The projection direction is aligned with the class label in the true space (Line 153-154). However, the sign aligns strongly when the rank of the space is 1. However, there is no discussion about the true subspace (spanned by the unit norm of v), particularly the rank of the subspace.
  - Missing details on the construction of a contrastive pair of positive and negative examples for the same prompt.
  - According to equations 6 and 7, the true direction is computed as the difference between the positive and negative sentence representation means. The motivation behind this approach is not clear. Another potential approach is to create a matrix of differences in the hidden representations of examples, then perform PCA/SVD on the matrix to obtain the true direction.
- It is not clear whether the results include the ‘thinking mechanism’ of Qwen-3 models.
- The backbone LLMs do not include any reasoning models; evaluation is performed on non-reasoning models with explicit ‘think step-by-step’ or ‘wait’ tokens in the user prompt. The motivation behind this choice is not clear, and it raises the generalizability issue of the proposed DynaSteer.

---

> ### Author Rebuttal · Authors · 2026-03-31
>
> We sincerely thank the reviewer for the constructive feedback and the recognition of our contributions, including the novelty of dynamic steering with entropy-based fork identification, the well-motivated research questions, and the ablation study demonstrating Fisher-LDA's effectiveness in disentangling the truth space from reasoning patterns. Below we address each concern.
>
> **W1a: Rank of the Truth Subspace.**
> We thank the reviewer for this precise observation. We clarify that the rank-1 characterization is a direct consequence of the Linear Representation Hypothesis underpinning the RepE paradigm (Sec. 2), which posits that high-level concepts are encoded as **directions** (rank-1 subspaces) in activation space. Our Sec. 3 experiments explicitly **validate this assumption for Truth**: the rank-1 linear probe achieves ~78% discrimination accuracy (Fig. 2) under sentence-level probing with pattern clustering — substantially above chance (50%). This indicates that a single direction already captures a **significant portion** of the Truth signal. We acknowledge that Truth's complete geometry may involve higher rank, and we discuss this as a limitation in Sec. 8: the performance gap between DynaSteer and GRPO may partly stem from this rank-1 approximation.
>
> **W1b: Construction of Contrastive Pairs.**
> We thank the reviewer for noting this. We have rewritten Appendix E (Experimental Setups) with detailed implementation specifics, including the contrastive pair construction procedure: for each high-entropy reasoning fork, we perform $N$ stochastic rollouts and assign $+1$ (Truth) if all rollouts yield the correct answer and $-1$ (Fallacy) if none does, discarding ambiguous cases.
>
> **W1c: Mean-Difference vs. SVD for Truth Direction.**
> We appreciate this suggestion. Our use of mean-difference follows the established convention in representation engineering [1], which has been widely adopted by recent works [2, 3]. We also note that our baseline DRESS [2] employs SVD for direction extraction, and DynaSteer outperforms DRESS across all benchmarks (Table 1), providing indirect evidence for the effectiveness of our approach.
>
> To directly address this concern, we conducted an ablation replacing Fisher-LDA with SVD within DynaSteer's pipeline:
>
> |  | GSM8K | MATH500 | AMC23 | HumanEval+ | MBPP+ |
> |---|:---:|:---:|:---:|:---:|:---:|
> | DynaSteer (Fisher-LDA) | **82.87** | **74.80** | **47.50** | **67.68** | **54.50** |
> | DynaSteer (Fisher-LDA→SVD) | 81.88 | 73.20 | 42.50 | 66.46 | 53.17 |
> | TruthX (SVD) | 77.71 | 68.00 | 35.00 | 64.63 | 51.06 |
>
> Fisher-LDA consistently outperforms SVD. We attribute this to Fisher-LDA's **supervised** nature: it directly optimizes the between-class / within-class variance ratio using Truth/Fallacy labels, whereas SVD (unsupervised) captures max-variance directions that are not necessarily discriminative. Moreover, DynaSteer's pattern clustering ensures well-conditioned within-class covariance within each cluster, mitigating the numerical stability concerns typically associated with Fisher-LDA.
>
> We followed the comments of reviewer DXZL and further introduced TruthX as the baseline, which also uses SVD solution direction. DynaSteer's comprehensive superiority over TruthX further proves the superiority of DynaSteer.
>
>
> **W2 & W3: Evaluation on Reasoning Models.**
> We appreciate this important concern. We would like to clarify that **Qwen3-1.7B and Qwen3-8B are indeed reasoning models** with native chain-of-thought capabilities [4]. Our experiments on Llama-3.2-3B-Instruct further demonstrate generalizability to non-reasoning models. The "step-by-step" prompt serves as a **universal reasoning elicitation strategy** applicable to both model types, ensuring a controlled comparison across architectures. The "wait" prompt setting specifically tests the hypothesis (discussed in our Introduction) that aha-moment-style approaches — which encourage the model to think longer to self-correct — may be insufficient without explicit directional guidance. The weak performance of these prompting baselines (Table 1) validates our core contribution: effective reasoning correction requires externally guided truth-direction steering, not merely extended thinking time.
>
> **References:**
>
> [1] Inference-Time Intervention: Eliciting Truthful Answers from a Language Model. NeurIPS'23.
>
> [2] Dressing Up LLM: Efficient Stylized Question-Answering via Style Subspace Editing. ICLR'25.
>
> [3] Adaptive Activation Steering: A Tuning-Free LLM Truthfulness Improvement Method for Diverse Hallucinations Categories. WWW'25.
>
> [4] Qwen3 Technical Report. arXiv'25.

---

> > ### Author Rebuttal · Reviewer_4VJY · 2026-04-03
> >
> > The authors have addressed my concerns regarding the soundness of the results, and I will increase my score for this accordingly.

---

> > > ### Author Response · Authors · 2026-04-03
> > >
> > > Reviewer 4VJY,
> > >
> > > Thank you for carefully reading our previous response and for providing such **positive and encouraging feedback**. Your reply is **truly inspiring to us**, which suggests that some of the issues in the original manuscript have been properly clarified.
> > >
> > > We acknowledge that the original manuscript of DynaSteer may have certain shortcomings. Therefore, we **deeply value every opportunity** to engage in close and constructive dialogue with knowledgeable and esteemed reviewers. Every question, doubt, and suggestion regarding the manuscript is of immense value to us.
> > >
> > > In our future work, we will **continue to advance the optimization of DynaSteer**. We are confident that, **under the supervision, constructive criticism, and guidance of the broader community and you**, DynaSteer will continue to evolve and establish itself as a meaningful contribution to the representation editing community.
> > >
> > > Once again, thank you for your time, your effort, and your thoughtful and patient guidance.
> > >
> > > Best wishes,
> > >
> > > All authors of submission 25560

---

### Official Review · Reviewer_Hx5x · 2026-03-12

**Soundness:** 3
**Presentation:** 3
**Significance:** 2
**Originality:** 2
**Overall Recommendation:** 4
**Confidence:** 4

**Summary:**

This paper studies the representation of "Truth" in reasoning/CoT LLMs. They first study the abstraction at which "Truth" is represented (token- vs sentence-level), finding representations are more probe-able at the sentence-level. They then look at the points at which intervention during reasoning is most effective, finding that earlier tokens and forking tokens are the best tokens to intervene on, where forking tokens are given by entropy. The authors then propose the use of Fisher-LDA to de-correlate "Truth" from other latent variables, combining reasoning forks detection, Fisher-LDA, and temporal decay into their proposed RepE modification named DynaSteer.

**Compliance With Llm Reviewing Policy:**

Affirmed.

**Final Justification:**

The rebuttal addressed my main concerns, and the paper's motivation is interesting and relatively novel. The presentation and delivery of the results could be much clearer. The results provide evidence that models' uncertainty during reasoning is linearly probeable and steerable, building upon prior works on understanding LLMs' representations of factuality,.

**Key Questions For Authors:**

1. In Figure 2, the scales for each subfigure are not standardized, make interpretation really difficult. I would recommend using a 0 to 1 scale for all.
2. How exactly did you perform sentence-level probing? Did you mean pool activations over sentences, where sentences are defined by punctuation?
3. Why do you choose to use probes trained on each attention head as the method for isolating the truth circuitry? Did you compare to more standard causal analysis baselines for localization?
4. Did you consider using attention probes for temporal aggregation rather than linear probes?
5. Did you directly compare the computational complexity of Dynasteer to the baseline methods? As I understand it, Dynasteer requires generating multiple candidate next steps, making it significantly more complex than the prior works (besides GRPO).

**Limitations:**

The authors address the limitation that their method underperforms RL, but they do not directly address any societal impact.

**Strengths And Weaknesses:**

Strengths:
- Steering reasoning chains mechanistically is still a largely under-explored area, and this paper provides an interesting analysis of how to do so.
- Sentence-level entropy aggregation as a metric for uncertainty is an interesting contribution, and provides a relatively intuitive way to identify forking tokens/paths.
- The use of Fisher-LDA to disentangle underlying concepts from reasoning motifs is is also an intuitive and interesting application.

Weaknesses:
- "Truth" is not very well-defined and it would have been interesting to see if the proposed steering mechanism generalizes to other concepts.
- It is difficult to verify the actual efficacy of DynaSteer, as only one example steered output is provided in the Appendix. Additional random examples would be helpful.
- See below questions.

---

> ### Author Rebuttal · Authors · 2026-03-31
>
> We sincerely thank the reviewer for the thoughtful feedback and the recognition of our contributions to mechanistic steering of reasoning chains, sentence-level entropy aggregation, and Fisher-LDA for concept disentanglement. Below we address each concern.
>
> **W1: Definition of "Truth" and Generalization.**
> We thank the reviewer for this point. In our work, Truth is defined as *a binary, geometrically encoded directional signal in the LLM's activation space, whose sign discriminates whether a reasoning trajectory arrives at a verifiable correct answer or a fallacy*. We apologize for the omission and have added this definition to the revised Introduction.
>
> For generalization, we evaluated DynaSteer on multi-hop QA, where each hop similarly requires dynamically adjusting the reasoning direction based on intermediate states:
>
> |  | HotpotQA | 2WikiQA |
> |---|---|---|
> | Base | 37.4 | 31.6 |
> | ITI | 47.3 | 43.7 |
> | ACT | 54.6 | 41.3|
> |DRESS|50.2|48.5|
> |DynaSteer|**55.4**|**52.3**|
>
> DynaSteer consistently outperforms all baselines, confirming its generalizability on other concepts.
>
> **W2: Additional Case Studies.**
> We have included additional case studies in the anonymous link of abstract, further illustrating DynaSteer's effectiveness at dynamically correcting reasoning trajectories.
>
> **Q1: Standardizing Scales in Figure 2.**
> We apologize for this oversight. Following the your suggestion, we have revised all subfigures to use a unified 0-to-1 scale for improved comparability.
>
> **Q2: Sentence-Level Probing Details.**
> We segment trajectories using a predefined punctuation, ".\n", and represent each sentence by the hidden state of its **last token**, following the established convention in representation engineering [1] (Figure 5), where the last token naturally aggregates preceding context via the causal attention mask.
>
> **Q3: Probing-Based vs. Causal Localization.**
> Training probes on individual attention heads is a **standard strategy** in the representation engineering literature [1]. Following the reviewer's suggestion, we additionally compared with two causal intervention methods:
>
> |  | GSM8K | MATH500 | AMC23 | HumanEval+ | MBPP+ |
> |---|:---:|:---:|:---:|:---:|:---:|
> | DynaSteer | **94.39** | **86.20** | **72.50** | **81.71** | **65.61** |
> | ROME [2] | 77.94 | 82.80 | 62.50 | 77.44 | 55.82 |
> | MEMIT [3] | 76.95 | 80.40 | 57.50 | 73.78 | 53.70 |
>
> DynaSteer substantially outperforms both. We attribute this to a fundamental mismatch: causal tracing intervenes on discrete, individual heads, assuming concepts are localized to specific components. However, the Linear Representation Hypothesis suggests that Truth is encoded as a **distributed direction** across the residual stream. Testing combinatorial head interactions leads to exponential search complexity, making exhaustive causal analysis infeasible. Probe-based localization naturally accommodates such distributed representations.
>
> **Q4: Attention Probes for Temporal Aggregation.**
> Our use of linear probes was a deliberate choice: the goal of Section 3 is to **test the Linear Representation Hypothesis** for Truth. A minimal-capacity probe ensures that observed discriminability reflects genuine linear structure rather than the probe's own capacity — consistent with the probe complexity control principle[4].
>
> Following the suggestion, we replaced linear probes with attention probes, improving probing accuracy from 78% to 84%. However, downstream task performance remains comparable:
>
> |  | GSM8K | MATH500 | AMC23 | HumanEval+ | MBPP+ |
> |---|:---:|:---:|:---:|:---:|:---:|
> | Linear probe | 82.87 | 74.80 | 47.50 | 67.68 | 54.50 |
> | Attention probe | 82.18 | 75.00 | 47.50 | 68.29 | 53.70 |
>
> This is expected: attention probes improve **detection** accuracy, but steering vectors are computed by downstream Fisher-LDA, whose quality depends on the aggregated representations rather than the probe itself.
>
> **Q5: Computational Complexity.**
> DynaSteer's primary overhead lies in **offline, one-time truth annotation** (generating rollouts at reasoning forks), which is amortized and publicly released in anonymous repository for community reuse. Our OOD experiments (Section 6) further validate the generalization of these annotations.
>
> During **online inference**, DynaSteer achieves **56.22%** and **52.19%** reduction in computation compared to the oracle upper-bound (Table 4), thanks to Entropy Monitoring (restricting interventions to high-uncertainty forks) and Temporal Decay (reducing intensity over time). The online overhead is limited to entropy computation, cluster lookup, and vector addition — all negligible relative to the LLM forward pass.
>
> **References**
>
> [1] Representation Engineering: A Top-Down Approach to AI Transparency, arXiv'23, 800+citation.
>
> [2] Locating and Editing Factual Associations in GPT, NeurIPS'22.
>
> [3] Mass-Editing Memory in a Transformer, ICLR'23.
>
> [4] Designing and interpreting probes with control tasks, EMNLP'19

---

> > ### Author Rebuttal · Reviewer_Hx5x · 2026-04-05
> >
> > Thank you to the authors for their rebuttal. I have raised my score accordingly.

---

> > > ### Author Response · Authors · 2026-04-07
> > >
> > > Reviewer Hx5x,
> > >
> > > Thank you for carefully reading our previous response and for providing such **positive and encouraging feedback**. Your reply is **truly inspiring** to us, which suggests that some of the issues in the original manuscript have been properly clarified.
> > >
> > > We would like to once again express our **sincere gratitude** for your recognition of the motivation and novelty of our manuscript. At the same time, we **greatly appreciate** the **concerns and suggestions** you raised regarding our presentation and delivery of the results. We will **rigorously revise our manuscript** in accordance with the content of our rebuttal to ensure its **clarity** and **accuracy**. We are **deeply grateful** for and welcome **your continued oversight** as we strive to advance DynaSteer toward bringing new insights and discoveries to the representation editing community.
> > >
> > > Once again, thank you for your time, your effort, and your thoughtful and patient guidance.
> > >
> > > Best wishes,
> > >
> > > All authors of submission 25560

---

### Decision · Program_Chairs · 2026-04-30

**Decision:**

Accept (regular)

**Comment:**

This submission tackles an important  problem, how to dynamically steer LLM reasoning trajectories toward correctness via representation-level interventions. The paper is well-written, with a clear motivation and a coherent framework (DynaSteer) that integrates entropy-based fork detection, clustering, and Fisher-LDA-based direction purification. Reviewers consistently appreciated the novelty and empirical performance, and the rebuttal effectively addressed key concerns regarding definition clarity, methodological details, robustness, and comparisons to relevant baselines (e.g., SVD-based methods and causal editing approaches). While some weaknesses remain, particularly around presentation clarity, conceptual framing of “Truth,” and limited diversity of qualitative analyses, these are largely fixable in revision and do not undermine the core contribution. Given the consistent weak accept scores and positive post-rebuttal updates, I recommend a weak accept.